# Homotopic contralesional excitation suppresses spontaneous circuit repair and global network reconnections following ischemic stroke

**Annie R Bice[1], Qingli Xiao[2], Justin Kong[3], Ping Yan[2], Zachary Pollack Rosenthal[4], Andrew W Kraft[2], Karen P Smith[2], Tadeusz Wieloch[5], Jin-Moo Lee[2], Joseph P Culver[6], Adam Q Bauer[1]***

[1]Department of Radiology, Washington University in St. Louis, Saint Louis, United States; [2]Department of Neurology, Washington University in St. Louis, Saint Louis, United States; [3]Department of Biology, Washington University in St. Louis, Saint Louis, United States; [4]Department of Neurology, Washington University School of Medicine, St. Louis, United States; [5]Department of Clinical Sciences, Lund University, Lund, Sweden; [6]Department of Radiology, Washington University in St. Louis, St. Louis, United States

**Abstract** Understanding circuit-level manipulations that affect the brain's capacity for plasticity will inform the design of targeted interventions that enhance recovery after stroke. Following stroke, increased contralesional activity (e.g. use of the unaffected limb) can negatively influence recovery, but it is unknown which specific neural connections exert this influence, and to what extent increased contralesional activity affects systems- and molecular-level biomarkers of recovery. Here, we combine optogenetic photostimulation with optical intrinsic signal imaging to examine how contralesional excitatory activity affects cortical remodeling after stroke in mice. Following photothrombosis of left primary somatosensory forepaw (S1FP) cortex, mice either recovered spontaneously or received chronic optogenetic excitation of right S1FP over the course of 4 weeks. Contralesional excitation suppressed perilesional S1FP remapping and was associated with abnormal patterns of stimulus-evoked activity in the unaffected limb. This maneuver also prevented the restoration of resting-state functional connectivity (RSFC) within the S1FP network, RSFC in several networks functionally distinct from somatomotor regions, and resulted in persistent limb-use asymmetry. In stimulated mice, perilesional tissue exhibited transcriptional changes in several genes relevant for recovery. Our results suggest that contralesional excitation impedes local and global circuit reconnection through suppression of cortical activity and several neuroplasticity-related genes after stroke, and highlight the importance of site selection for targeted therapeutic interventions after focal ischemia.

## Editor's evaluation

Overall, Bice et al., present new work using optogenetics-based stimulation to test how this affects stroke recovery in mice. The study provides interesting evidence that this stimulation may be harmful, and not helpful, and opens new avenues for both basic and therapeutic research.

## Introduction

Stroke causes direct structural damage to local brain circuits and indirect functional damage to global networks resulting in behavioral deficits spanning multiple domains (*Murphy and Corbett, 2009*; *Baldassarre et al., 2016*). In the months following stroke, functional magnetic resonance imaging

*For correspondence:
aqbauer@wustl.edu

Competing interest: The authors declare that no competing interests exist.

(fMRI) studies have shown that local circuits lost to infarction remodel in peri-infarct cortex (*Cramer and Riley, 2008*). This process, termed 'remapping', appears to be tightly focused to perilesional regions in patients exhibiting good recovery, and suggests that peri-infarct cortex may take over the function of brain regions lost to stroke (*Brown et al., 2009*; *Dijkhuizen et al., 2001*; *Nudo et al., 1996*). Similar studies examining brain-wide patterns of synchronized, resting-state, hemodynamic activity after stroke have shown that global patterns of resting-state functional connectivity (RSFC) are also altered (*Posner et al., 1988*; *Felleman and Van Essen, 1991*). Shortly after ischemic stroke, disruption of interhemispheric homotopic RSFC predicts poor motor and attentional recovery (*Carter et al., 2012*; *He et al., 2007*), and in rats, restoration of homotopic RSFC correlates with improved behavioral performance (*van Meer et al., 2010*). Similarly, we have shown in mice that functional disruption correlates with infarct size, regional RSFC disruption, the degree of remapping, and behavior (*Kraft et al., 2018*, *Bauer et al., 2014*). While functional neuroimaging studies consistently demonstrate local and global changes in functional brain organization after stroke, it is unknown how these processes inter-relate to support recovery of function.

Activity in brain regions functionally connected to perilesional tissue can differentially affect recovery potential. In monkeys, the use of the affected limb is required for remapping (*Nudo et al., 1996*), and conversely, overactivation of the contralateral hemisphere through the use of the 'good limb' is associated with poorer clinical outcome (*Turton et al., 1996*; *Feydy et al., 2002*). Additionally, there is substantial indirect evidence that homotopic, interhemispheric connections may directly impact recovery after stroke. Stroke patients with impaired hand movement experience abnormally high interhemispheric inhibition exerted from intact contralesional motor cortex to perilesional motor cortex, with the degree of inhibition increasing with deficit severity (*Nudo, 2007*). It is thought that an equilibrium between excitation/inhibition across hemispheres may be important for normal unilateral function (*Bloom and Hynd, 2005*). If the interhemispheric balance is disrupted (e.g. by stroke) local ipsilesional inhibitory influences from distant connections may worsen functional impairment. Prior human studies have primarily used non-invasive brain stimulation techniques (e.g. repetitive transcranial magnetic stimulation [rTMS]) to investigate the role of contralateral activity on the brain (*Nowak et al., 2010*). However, treatment efficacy using these methods is extremely varied (*Takeuchi et al., 2005*; *Nowak et al., 2008*; *Adeyemo et al., 2012*; *Hao et al., 2013*) and are limited by imprecision and indiscriminate activation or inhibition of all cell types near the stimulated site. Thus, it has been difficult to systematically examine the influence of specific neuronal populations and connection pathways important for recovery in human stroke patients (*Nudo, 2007*; *Takatsuru et al., 2013*; *Takatsuru et al., 2014*). Optogenetics is a powerful approach for increasing the specificity of paradigms designed to modulate local and distant neural activity after experimental stroke (*Cheng et al., 2014*; *Wahl et al., 2017*; *Song et al., 2017*; *Tennant et al., 2017*; *Shah et al., 2017*). The aim of this study was to determine if local circuits lost due to stroke in mice remap into perilesional cortex and reconnect with global functional circuits (i.e. resting-state networks), and if this process could be manipulated (spatially and temporally) by contralesional, homotopic excitation. Given that contralateral limb use suppresses recovery, we hypothesized that optogenetically exciting the contralesional cortex would likewise inhibit recovery. To mechanistically examine whether increased contralesional homotopic excitatory activity worsens recovery, we chose to lesion somatosensory forepaw cortex because this modality is amenable to tracking cortical remapping of sensory evoked responses in tandem with local and global network activity. Our results suggest that limited, focal excitation of contralesional cortex impedes perilesional remapping and restoration of RSFC within the somatomotor network. Furthermore, this maneuver profoundly altered more global patterns of cortical activity outside of the targeted circuit as well as recovery of the brain's global functional connectome. These observations were associated with regulation of several molecular pathways relevant for recovery after injury.

## Results
### Study design

In this study, we used optogenetic targeting in conjunction with optical intrinsic signal imaging (OISI) in CamK2a-ChR2 mice (see Methods) to examine the effects of contralesional excitation on local and global cortical remodeling after focal ischemia. Our experimental design is illustrated in *Figure 1*. Note that all cortical regions and manipulations are in reference to hemisphere (instead of limb to

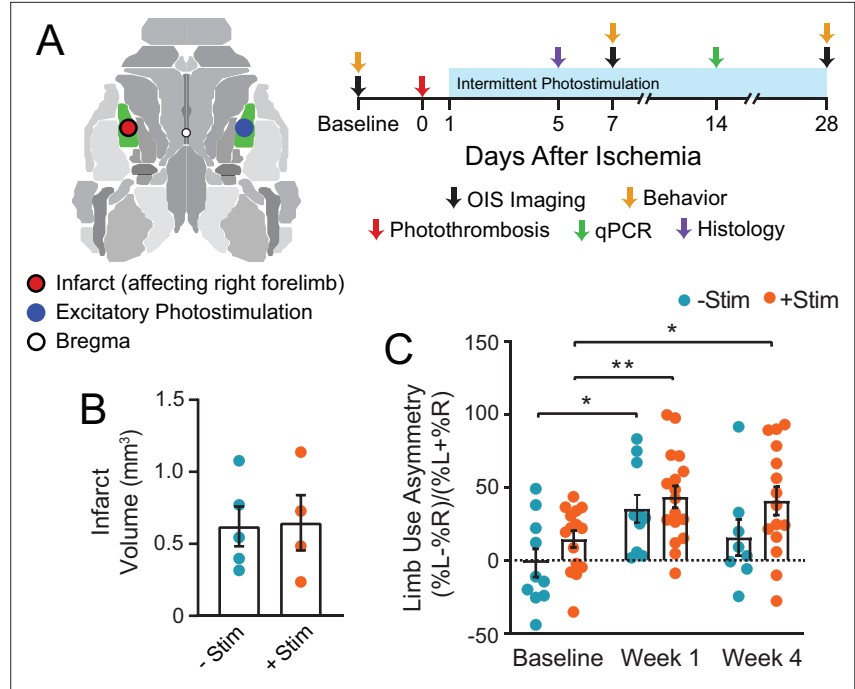

**Figure 1.** Graphical illustration of experiment and timeline. (**A**) Photothrombosis was delivered to primary somatosensory forepaw cortex (S1FP, shown in green) in the left hemisphere. Beginning on day 1 after focal ischemia, a subset of mice received interventional optogenetic photostimulation of homotopic S1FP in the right hemisphere for 3 min/day. Treatment was given for 5 consecutive days/week for 4 weeks. Optical intrinsic signal imaging of stimulus evoked activity (following electrical stimulation of the forepaws or optogenetic photostimulation of S1FP) and resting-state activity occurred before, and 1 and 4 weeks after stroke. Infarct volume and mRNA expression were characterized in a subset of mice from both groups at 5 and 14 days after photothrombosis. (**B**) Infarct volume characterized 5 days post photothrombosis for −Stim (n=5) and +Stim (n=4) groups were statistically equivalent. (**C**) Limb use as measured by cylinder rearing test in −Stim (n=15) and +Stim (n=20) groups. Symmetrical limb use was observed at baseline time points in all mice. Photothrombosis resulted in use asymmetry due to decreased use of right forelimb within the first week of both groups. The −Stim mice demonstrated significant improvement by week 4 while +Stim mice exhibited sustained use asymmetry at 4 weeks. Linear mixed-effects model (LMM) revealed a main effect of time ($F_{1.77,39}$=8.7, p=0.0011). Post hoc tests were performed as t-tests assuming unequal group variance and corrected for multiple comparisons using false discovery rate correction. *=p<0.05; **=p<0.01; ***=p<0.001 compared to baseline time point. Statistical analyses were performed in Graph Pad Prism 8. All data reported as mean ± S.E.

which a brain region is mapped). We performed photothrombosis targeted to left primary somatosensory forepaw (S1FP) cortex. A randomly selected subset of mice received chronic, intermittent contralesional excitation of right S1FP (+Stim) beginning day 1 after photothrombosis and continued for 5 consecutive days/week over 4 weeks. The remaining mice recovered spontaneously (−Stim). Assessments of behavior (via cylinder rearing) and brain function (via OISI) occurred before, and 1 and 4 weeks after stroke. A sham group controlled for the effects of chronic photostimulation of right S1FP cortex in the absence of stroke and were imaged at the same time points as the stroke groups. Infarct volume and mRNA expression were characterized in a subset of mice from both stroke groups on days 5 and 14 post stroke, respectively.

## Contralesional excitation inhibits behavioral recovery and cortical remapping and after stroke

We implemented a photostimulation protocol previously reported to positively affect recovery when chronically applied to ipsilesional tissue after stroke (*Cheng et al., 2014*). To ensure that our stimulation paradigm did not exacerbate the degree of initial injury, average infarct volume (in mm³) was quantified 5 days after photothrombosis. Both groups had statistically equivalent lesion sizes (t(5.7)=0.11, p=0.92, *Figure 1B*). Using the cylinder rearing test, symmetrical limb use was observed at baseline in

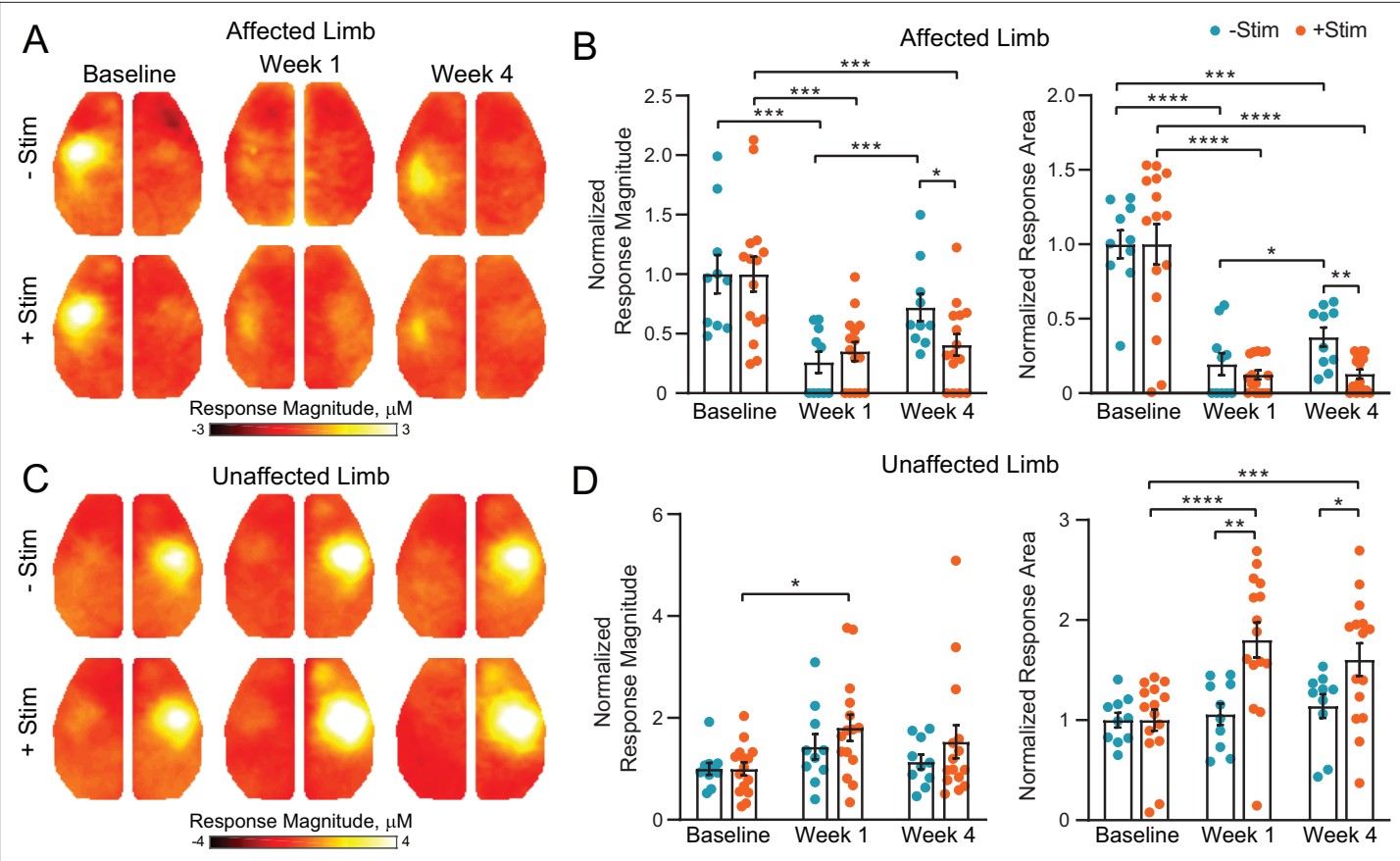

**Figure 2.** Contralesional activity inhibits cortical remapping after stroke. (**A**) In the affected (right) limb, stimulus evoked responses following electrical forepaw stimulation were significantly reduced 1 week after stroke in both groups. Mice recovering spontaneously (–Stim, n=10, top row) exhibited robust activations at 4 weeks, while responses in stimulated mice (+Stim, n=15) were comparable to week 1 responses. (**B**) Quantification of evoked response magnitude and area in the ipsilesional hemisphere. The 2-way repeated measures ANOVA (rmANOVA) during recovery revealed a significant group × time interaction ($F_{1,23}=5.12$; p=0.033) for evoked response magnitude and a trending group × time interaction ($F_{1,23}=4.0$; p=0.057) in response area. Main effects of group ($F_{1,23}=9.5$; p=0.005) and time ($F_{1,23}=4.5$; p=0.045) were observed for response area while magnitude also exhibited a main effect of time ($F_{1,23}=8.32$; p=0.008). (**C**) In the unaffected (left) limb, –Stim mice exhibited a trend toward increased activity at week 1 that subsided by week 4. In +Stim mice, evoked responses were significantly elevated at weeks 1 and 4 compared to baseline. (**D**) Quantification of evoked response magnitude and area in the contralesional hemisphere. Over the duration of the experiment, 2-way rmANOVA revealed a significant group x time interaction ($F_{2,46}=6.02$; p=0.0048), as well as a main effect of time ($F_{1.7,39}=9.34$; p=0.0009) and group ($F_{1,23}=6.2$; p=0.02) for evoked response area. A main effect of time ($F_{1.8,41}=6.43$; p=0.0051) was observed for response magnitude. Post hoc tests were performed as t-tests assuming unequal group variance and corrected for multiple comparisons using false discovery rate correction. All analysis performed using total hemoglobin as contrast. Statistical analyses were performed in Graph Pad Prism 8. *=p<0.05; **=p<0.01; ***=p<0.001; ****=p< 0.0001.

all mice (*Figure 1C*), with no group differences before stroke (t(15.3)=1.4, p=0.9). Photothrombosis resulted in decreased use of right forelimb within the first week of both groups (–Stim mice: t(9)=3.1, p=0.018 and +Stim mice: t(14)=3.5, p=0.001). Similar performance deficits were observed in both groups at week 1 (t(19)=0.68, p=0.5). At week 4, performance in –Stim mice was comparable to baseline (t(7)=1.4, p=0.19), while +Stim mice continued to exhibit sustained use asymmetry compared to baseline (t(12)=2.5, p=0.028).

Recovery of sensorimotor function is associated with the formation of new cortical representations of the affected modality (remapping) (*Cramer et al., 1997*; *Kraft et al., 2018*). Use of the unaffected limb is thought to be detrimental to recovery, we therefore hypothesized that contralesional excitation would negatively impact remapping (explored in *Figures 2 and 3*). Because evoked responses at early time points post stroke can be difficult to detect, group-averaged maps of peak responses at baseline were thresholded at 75% of maximum to define a threshold for what constituted a response for all mice at all time points. For *Figure 2*, the maximum response above this threshold was used for activation magnitude calculations, and the number of pixels above this threshold was used for the area

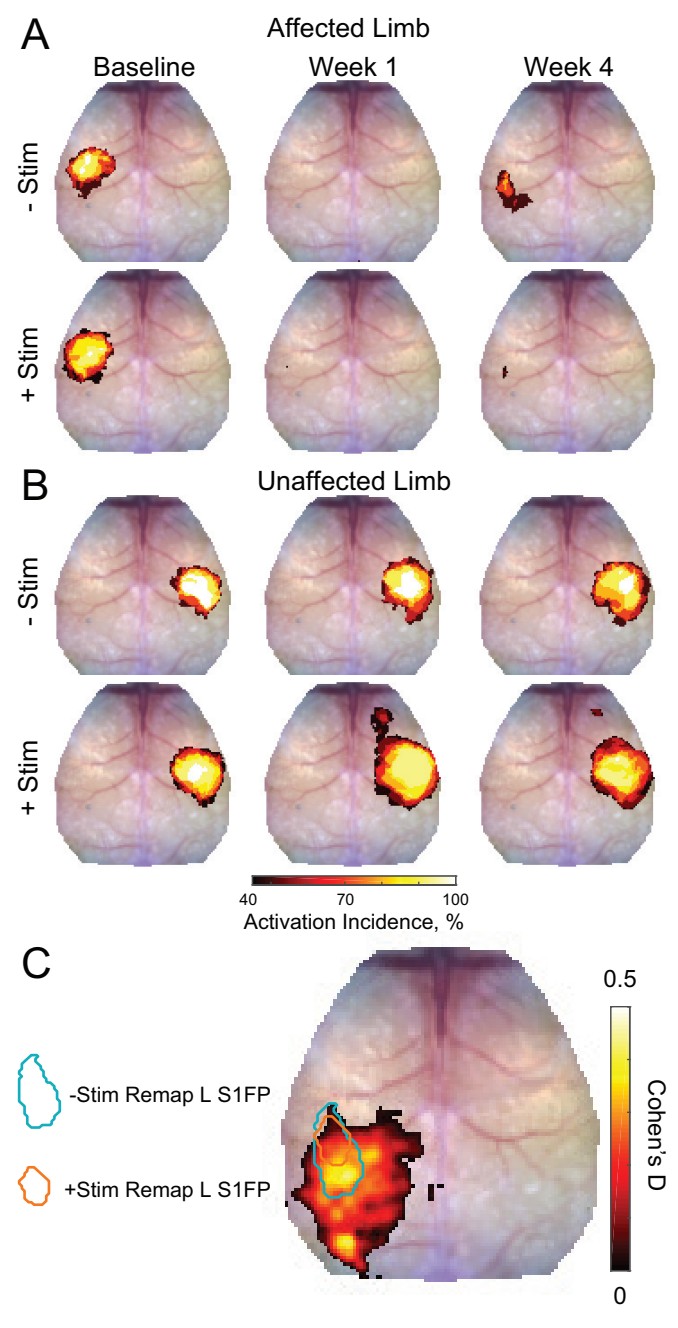

**Figure 3.** Activation incidence and effect size of forepaw remapping maps of activation incidence for evoked responses reported in *Figure 2* in the (**A**) affected and (**B**) unaffected limb of each group (–Stim [n=10] and +Stim [n=15]). 100% of mice produced robust responses at baseline in both limbs. (A) About 1 week after stroke, both groups exhibited significantly reduced activations in the affected limb, with at most 50% of mice having detectible responses in limited regions of cortex. By 4 weeks after stroke, 80% of –Stim mice exhibited reproducible responses in large regions of somatosensory cortex, posterior to the original forepaw region. Conversely, only 67% of +Stim mice had detectible responses that were smaller in total area. (**B**) In the unaffected limb, responses in the –Stim mice were largely similar across time points. However, responses in +Stim mice were generally larger (in line with *Figure 2*) and exhibited increased activity in forepaw motor cortex compared to the baseline time point. (**C**) Map of effect size showing main topographical differences in remapping between groups at 4 weeks following electrical stimulation of the affected forepaw. Cohen's D was calculated using mouse-level T-maps (see main text) as –Stim minus +Stim so that an effect would manifest as a positive value. Contours of original and remapped forepaw response shown for reference. Statistical analyses were performed in MATLAB (2018a).

*Figure 3 continued on next page*

*Figure 3 continued*

The online version of this article includes the following figure supplement(s) for figure 3:

**Figure supplement 1.** Focal excitatory stimulation influences stimulus evoked responses in healthy mice.

calculations. All magnitude and area values were then normalized by the mean baseline values. For *Figure 3*, activation incidence images (i.e. the percentage of mice that exhibited an evoked response) were created by binarizing all pixels above the 75% threshold for each mouse and time point.

At baseline, electrical stimulation of the right forepaw (affected limb) resulted in consistent activation of left S1FP cortex in 100% of mice in both groups (*Figures 2A and 3A*). Photothrombosis targeted to left S1FP resulted in >70% loss of evoked response magnitude in the first week following injury in both groups (–Stim: t(9)=4.6, p=0.001; +Stim: t(14)=4.4, p<0.001), and at least an 80% reduction in response area in both groups (–Stim: t(9)=8.6, p<0.0001; +Stim: t(14)=6.5, p<0.0001, *Figure 2B*). Responses from the affected limb reappeared in 80% of the mice in the –Stim group (*Figure 3A*) by 4 weeks, and exhibited magnitudes statistically indistinguishable (t(9)=1.3, p=0.22) from baseline responses (*Figure 2B*). These new cortical representations in –Stim mice were focused to regions lateral and posterior to the infarct (near barrel cortex) (*Figure 2A*, *Figure 3A*), significantly larger than those at week 1 (t(9)=2.3, p=0.045), but remained smaller than baseline representations (t(9)=6.0, p=0.0002, *Figure 2B*). In +Stim mice, only 67% of mice had detectable responses at week 4 (*Figure 3A*). Week 4 responses in +Stim mice were 60% smaller in magnitude (t(14)=4.6, p=0.0004) and 87% smaller in area (t(14)=7.57, p<0.0001) than those at baseline and were indistinguishable from response area (t(14)=0.1, p=0.92) and magnitude (t(14)=0.5, p=0.63) observed at week 1 (*Figure 2B*).

Left forepaw (unaffected limb) stimulation resulted in consistent activation of right S1FP cortex in 100% of the mice in both groups prior to photothrombosis (*Figure 2C*, *Figure 3B*). In +Stim mice, evoked responses from the unaffected limb at week 1 were 81% larger than baseline responses (t(14)=3.5, p=0.003, *Figure 2D*). Activation magnitude in –Stim mice trended (t(9)=1.7, p=0.1) toward larger responses at week 1, but returned to baseline levels by week 4 (t(9)=1.4, p=0.2). In +Stim mice, activation area was 80% larger at week 1 (t(14)=6.48, p<0.0001) and 60% larger at week 4 (t(14)=4.12, p=0.001) than those at baseline. Furthermore, stimulation of the unaffected limb in +Stim mice activated forepaw representations in contralesional motor cortex in 80% of mice that were not observed at baseline (*Figure 3B*). Post stroke activation area in –Stim mice did not significantly differ from baseline at either time point (week 1: t(9)=0.6, p=0.66; week 4: t(9)=1.3, p=0.21, *Figure 2D*), nor elicit responses in contralesional motor cortex (*Figure 3B*).

To further examine the effect of contralesional stimulation on group-wise cortical remapping, T-maps were calculated for each mouse for all imaging sessions of evoked activity at 4 weeks following electrical stimulation of the affected forepaw. From these T-maps, a map of Cohen's D was calculated as –Stim minus +Stim so that an effect would manifest as a positive value (*Figure 3C*). Computed differences between groups included the region of forepaw remapping in the ipsilesional hemisphere dominated by –Stim mice. Peak Cohen's D in the region of remapping was 0.44, an intermediate effect, indicating that –Stim mice exhibit a larger, more active portion of remapped cortex compared to +Stim mice.

## Focal excitatory stimulation in control mice results in increased evoked responses of right and left S1FP

To determine if the effects of contralesional stimulation were conditional on focal ischemia, stimulus evoked responses were characterized in mice experiencing chronic excitatory stimulation of right S1FP in the absence of photothrombosis (*Figure 3—figure supplement 1*). Robust evoked activity was observed in all control mice and time points (*Figure 3—figure supplement 1A*). Interestingly, by 4 weeks, this group also exhibited increased evoked activity in forepaw representations of motor cortex, similar to contralesional responses observed in +Stim mice after stroke. In control mice, stimulus evoked responses in the right hemisphere exhibited larger response area at 1 week (t(9)=4.4, p=0.002) and 4 weeks (t(8)=6.1, p=0.0003) compared to baseline, and between weeks 1 and 4 (t(8)=3.96, p=0.004) (*Figure 3—figure supplement 1B* top). Response magnitude in the right hemisphere was larger at week 4 compared to week 1 (t(8)=3.33, p=0.01) (*Figure 3—figure supplement 1B* top). In the left hemisphere, response area was significantly increased at 4 weeks compared to

baseline (t(6)=4.43, p=0.004) and at 4 weeks compared to 1 week (t(7)=3.02, p=0.02) (*Figure 3—figure supplement 1B* bottom). Response magnitude did not significantly change over time, but trended toward a larger response between weeks 1 and 4 (t(7)=2.57, uncorrected p=0.037).

## Contralesional stimulation inhibits recovery of somatomotor forepaw circuitry

Both remapping (*Brown et al., 2009*; *Kraft et al., 2018*) and the formation of new homotopic functional connections (*van Meer et al., 2010*; *Carter et al., 2012*) are associated with better recovery following stroke. Having observed differential remapping in +Stim and –Stim groups, we examined how contralesional excitation affected reintegration of local S1FP circuitry into the somatomotor resting-state network. The regions of interest (ROIs) for RSFC analysis were defined by stimulus evoked responses at baseline and week 4 (see Methods): original S1FP representations in both hemispheres, new (remapped) S1FP representations in the lesioned hemisphere, and forepaw representations in contralesional motor cortex, primary motor forepaw cortex (M1FP), (*Figure 4A*). Group-averaged S1FP RSFC maps at baseline reveal positive ipsilateral RSFC with sensory and motor regions, as well as bilateral RSFC with original S1FP in both groups (*Figure 4B*). About 1 week after photothrombosis, bilateral RSFC between original ipsilesional and contralesional S1FP in both groups is significantly reduced (–Stim: t(9)=6.2, p=0.0002; +Stim: t(14)=3.1, p=0.008; *Figure 4B and C*). Furthermore, at week 1, RSFC between contralesional S1FP and the region of future ipsilesional remapping has not been established in either group (–Stim: t(9)=4.6, p=0.0014; +Stim: t(14)=3.5, p=0.0036) (*Figure 4B and D*, *Figure 4—figure supplement 1A*). However, in –Stim mice 4 weeks after stroke, new 'homotopic' functional connections between contralesional S1FP and remapped ipsilesional S1FP were observed having RSFC strength equivalent to baseline values (t(9)=0.29, p=0.78). Conversely, RSFC between contralesional S1FP and remapped S1FP at 4 weeks in the +Stim group was significantly reduced from baseline (t(14)=2.2; p=0.042) and indistinguishable from week 1 (t(14)=0.28, p=0.79; *Figure 4B and D*, *Figure 4—figure supplement 1*). Furthermore, significant disruption of intrahemispheric, ipsilesional RSFC between remapped S1FP and motor persisted at 1 and 4 weeks in +Stim mice (1 week: t(14)=2.9, p=0.011; 4 weeks: t(14)=3.8, p=0.002) (*Figure 4—figure supplement 1B*).

As reported in *Figure 2C and D*, electrical stimulation of the unaffected forepaw resulted in a trend toward (–Stim) or significant increase in (+Stim) stimulus evoked activity in contralesional S1FP and M1FP following stroke. We next examined if increased contralesional somatomotor activity was associated with increased intrahemispheric RSFC between these regions (*Figure 4E*). In +Stim mice, intrahemispheric RSFC between sensory and motor cortices did not appreciably change over time from baseline values, despite exhibiting increased area and magnitude in these regions after photothrombosis. At week 4, –Stim mice exhibited higher intrahemispheric RSFC between sensory and motor regions compared to +Stim mice (t(20.5)=3.01, p=0.0068) suggesting the involvement of the contralesional hemisphere during spontaneous recovery. At week 1, +Stim mice demonstrated a small but significant (t(20.4)=2.4, p=0.028) increase in intrahemispheric RSFC compared to –Stim mice. However, this increase did not persist in +Stim mice over time. In control mice, photostimulation did not significantly alter homotopic S1FP RSFC (*Figure 4—figure supplement 2A,B*) or intrahemispheric RFC between right S1FP and right M1FP (*Figure 4—figure supplement 2C*). Together, these results suggest that more complete cortical remapping is associated with a return to more normalized patterns of interhemispheric, homotopic RSFC in the affected circuit. Furthermore, larger patterns of activation in the unaffected hemisphere are not indicative of increased intrahemispheric RSFC within that hemisphere.

## Global network interactions after stroke are suppressed by homotopic contralesional stimulation

Functional network disruption following focal ischemia extends outside of the lesioned territory to brain regions remote from the infarct (*Griffis et al., 2019*). Brain networks returning toward more normal patterns of intrinsic organization after stroke (i.e. restored homotopic RSFC) appear to support better behavioral performance (*Carter et al., 2012*; *He et al., 2007*). Because +Stim mice exhibited sustained behavioral deficits and lower S1FP RSFC at week 4 compared to –Stim mice, we hypothesized that contralesional focal excitatory stimulation also altered the degree to which global RSFC renormalized in +Stim mice. Whole cortex correlation matrices were generated for all

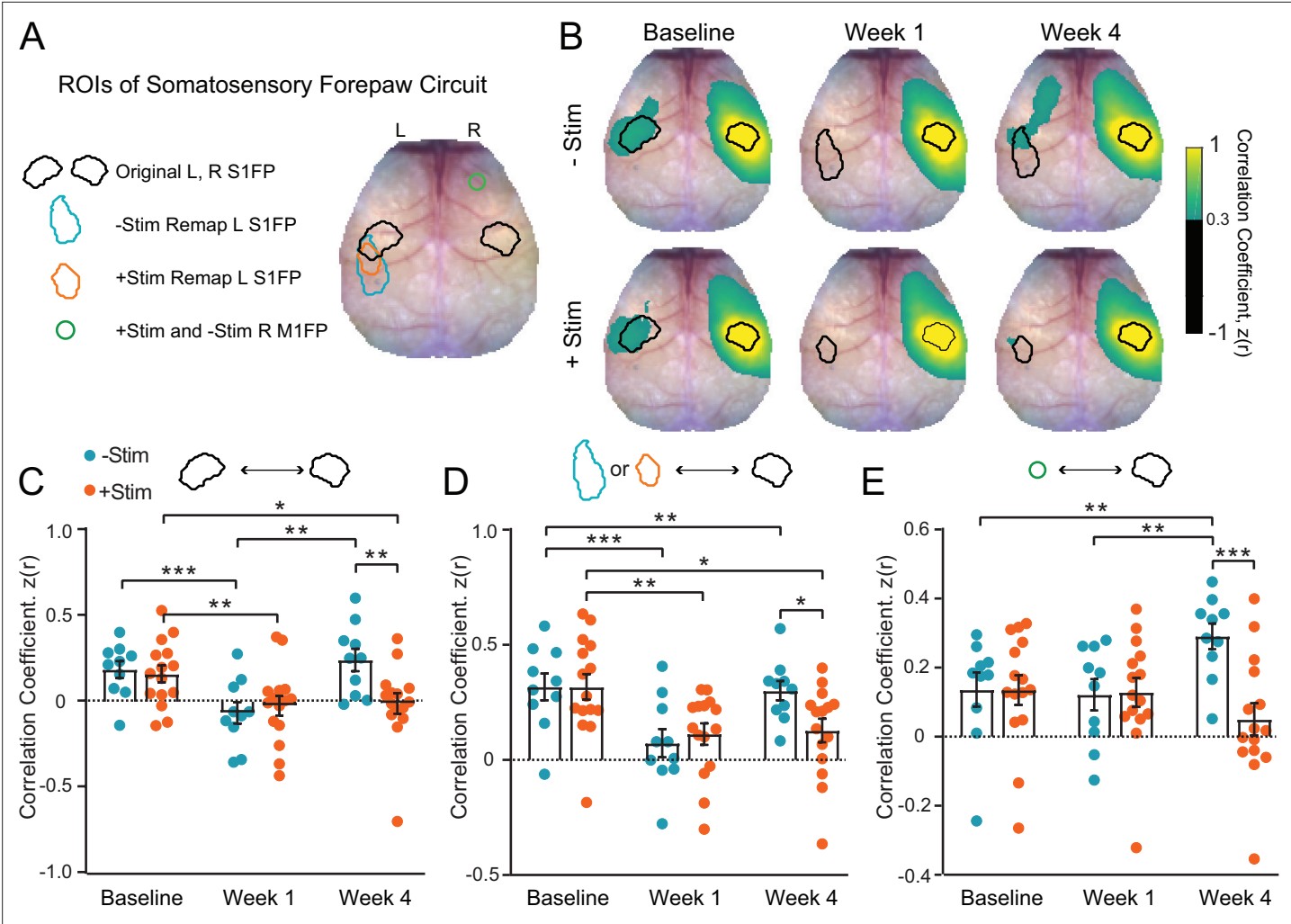

**Figure 4.** Contralesional stimulation inhibits recovery of somatomotor forepaw circuitry (**A**) regions of interest (ROIs) overlaid on a representative white light image of the dorsal mouse skull. ROIs were determined from evoked responses at baseline (original primary somatosensory forepaw [S1FP] in the left or right hemispheres) or at 4 weeks after stroke (remapped left S1FP). Responses at 4 weeks contained increased activity in right M1FP and included as an ROI for S1FP resting-state functional connectivity (RSFC) analysis. (**B**) Group-averaged maps of RSFC for the original right S1FP ROI in both –Stim (n=10) and +Stim (n=15) groups reveal strong ipsilateral connectivity with sensory and motor regions, as well as bilateral RSFC with left forepaw cortex. Similar deficits are observed in both groups 1 week after stroke. Bilateral RSFC in both groups is nearly ablated while ipsilateral connectivity remains relatively preserved. By 4 Weeks, RSFC in right S1FP in the –Stim group exhibits new functional connections in perilesional somatosensory regions, including remapped left S1FP as well as more anterior motor cortices. Interhemispheric RSFC of right S1FP in the +Stim group remains largely absent by 4 weeks. (**C–E**) Quantification of RSFC over time in ROIs depicted in (**A**). (**C**) Homotopic RSFC in original S1FP is significantly reduced in both groups after stoke, and only recovers in –Stim mice: 2-way repeated measures ANOVA (rmANOVA) revealed a significant group × time interaction ($F_{2,46}$=5.3, p=0.009) and a main effect of time ($F_{1.7,40}$=11.2, p=0.0003). (**D**) 'Homotopic' RSFC with remapped forepaw regions is reinstated by 4 weeks after stroke in –Stim mice only: during recovery, 2-way rmANOVA revealed a significant group × time interaction ($F_{1,23}$=5.9, p=0.023) and a main effect of time ($F_{1,23}$=7.8, p=0.01). (**E**) Ipsilateral RSFC between right S1FP and right M1FP. Despite exhibiting increased activity in R right M1FP during left forepaw stimulation, +Stim mice do not exhibit increased ipsilateral RSFC between right M1FP and right S1FP. 2-way rmANOVA revealed a significant group × time interaction ($F_{2,46}$=10.52, p=0.0002). All post hoc tests were performed as t-tests assuming unequal group variance and corrected for multiple comparisons using false discovery rate correction. Statistical analyses were performed in Graph Pad Prism 8. *=p<0.05; **=p<0.01; ***=p<0.001; ****=p<0.0001.

The online version of this article includes the following figure supplement(s) for figure 4:

**Figure supplement 1.** Resting-state functional connectivity (RSFC) in remapped primary somatosensory forepaw (S1FP).

**Figure supplement 2.** Focal excitatory stimulation in control mice does not affect resting-state functional connectivity (RSFC) in the somatomotor circuit.

pairwise comparisons within our field-of-view for both groups at week 4 (*Figure 5A and B*). Matrices are grouped by functional assignment then organized by hemisphere (left, ipsilesional; right contrale-sional). Qualitatively, –Stim mice (*Figure 5A*) exhibit stronger homotopic RSFC (off-diagonal elements) compared to +Stim mice (*Figure 5B*). Furthermore, ipsilateral anticorrelations between anterior-posterior brain regions are also more pronounced, for example between ipsilesional motor and visual regions, in –Stim mice compared to +Stim mice. Group-wise correlation differences at week 4 were calculated as –Stim minus +Stim (*Figure 5C*). Compared to +Stim mice, –Stim mice exhibit increased interhemispheric RSFC between ipsilesional sensory and motor cortex (e.g. Box 1, *Figure 5C*), homotopic RSFC within the somatosensory and motor networks (e.g. Box 2, *Figure 5C*), as well as increased intra- and inter-hemispheric RSFC within visual and parietal cortex (e.g. Box 3, *Figure 5C*). Additionally, mice recovering spontaneously exhibit increased anticorrelations between anterior (e.g. motor and sensory) and posterior (visual) cortices (Box 4, *Figure 5C*).

Maps of the most salient group differences in RSFC were created using unbiased spatial principal component analysis (PCA) of the group-level, week 4 correlation difference matrix (*Figure 5D and E*). PC1 reveals the topography associated with the largest RSFC differences between groups. This map, which explains 52% of the inter-group variance at week 4, includes large portions of motor and somatosensory cortex (reds) and an island of connectivity within the remapped forepaw regions. Pronounced anticorrelations (blues) can also be observed between posterior visual/retrosplenial regions and anterior sensorimotor regions. PC2, which explains 18% of the variance between groups at week 4, shows that perilesional and contralesional sensory, medial motor, cingulate, and more distant parietal regions also exhibit increased RSFC in –Stim mice compared to +Stim mice. The resulting eigenspectrum after PCA of the week 4 correlation difference matrix reveals that, in total, these observations explain 70% of the variance between groups (PC1: 52%; PC2: 18%; *Figure 5E*).

In order to generate unbiased ROIs for probing evolving global RSFC after stroke, the first 2 PCs shown in *Figure 5D* were each averaged across midline, smoothed, and thresholded at a confidence interval of 85% for both positive and negative values. This procedure produced 7 PCA-derived ROIs in each hemisphere corresponding to regions overlapping with primary and secondary motor (M1 and M2), primary sensory hindpaw/forepaw (S1hp/fp), retrosplenial (Ret), anterior and posterior primary visual (V1a and V1p), and secondary visual (V2) cortex (*Figure 5F*, *Figure 5—figure supplement 1A*). For each mouse and group, pixel time traces were averaged within each ROI and correlated to generate regional RSFC matrices at each time point for all pairwise comparisons(*Figure 5—figure supplement 1B*). The 2-way repeated measures ANOVA (rmANOVA) applied to each matrix element over time shows brain-wide differences in regional RSFC over time between groups (*Figure 5G*). The p-values are shown for all elements having a significant group × time interaction (uncorrected p<0.05). Regions closest to direct injury (i.e. S1HP/FP) exhibited significant differences in RSFC with visual brain regions. Furthermore, RSFC within and across regions distant to the lesion was also significantly affected, for example, inter- and intrahemispheric RSFC within visual and retrosplenial cortices, as well as RSFC between visual and motor regions across hemispheres.

Global network recovery was determined in each group by subtracting the magnitude of ROI-based RSFC at week 1 from those at week 4 (*Figure 5—figure supplement 1B,C*). Significant differences (FDR corrected p<0.05) were visualized as a stick and ball network diagram on the mouse skull with line thickness indicating connection strength (*Figure 5H*, left). Comparing recovery matrices of –Stim mice vs +Stim mice revealed that chronic excitation after stroke predominantly suppresses restoration of global RSFC (blue lines, *Figure 5H*, left). Specifically, +Stim mice exhibited significantly suppressed recovery between nodes connected to the lesion and other regions (left S1hp/fp and left visual; left M2 and left retrosplenial), homotopic nodes distant to the site of direct injury (e.g. left and right visual cortices), and across hemispheres (e.g. right motor and left visual). The effects of chronic stimulation on evolving global RSFC appear to depend on the post stroke environment. In control mice, chronic right S1FP stimulation did not significantly alter RSFC between any ROI pairs over the course of 4 weeks (*Figure 5H*, right).

## Recovery of functional connection density is suppressed by focal contralesional activity

In addition to mapping pairwise RSFC differences across groups, we also examined the number of functional connections (node degree) exhibited by each pixel over the brain (*Figure 5—figure*

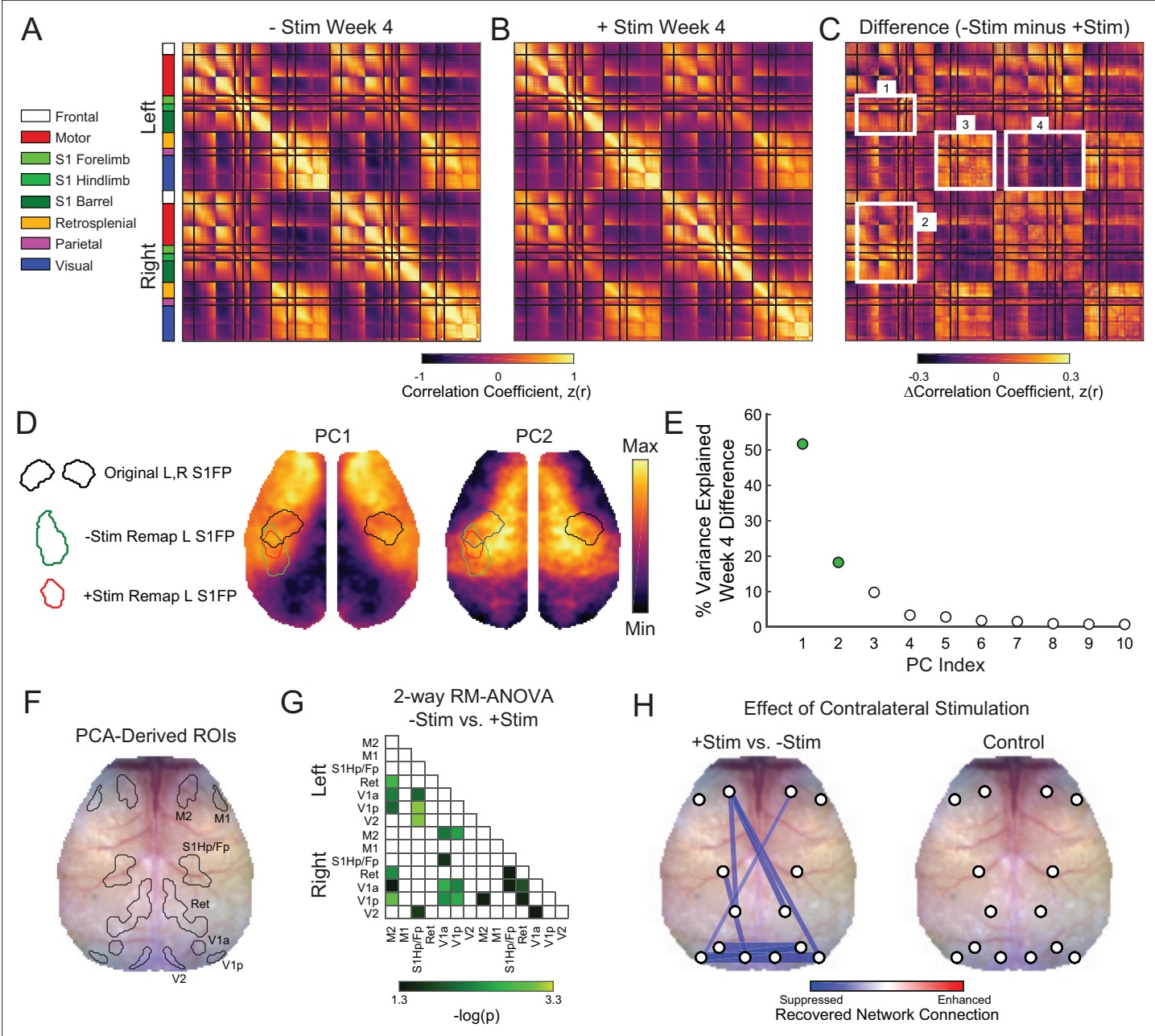

**Figure 5.** Global network interactions after stroke are suppressed by increased contralesional forepaw activity whole cortex correlation matrices for (**A**) –Stim (n=10) and (**B**) +Stim (n=15) groups 4 weeks after photothrombosis. Matrices are grouped by functional assignment then organized by hemisphere (left, ipsilesional; right, contralesional). (**C**) Resting-state functional connectivity (RSFC) difference matrix (–Stim minus +Stim) shows the group-averaged correlation differences between experimental groups. Notice increased intra- (box 1) and inter-hemispheric (**C**) RSFC within the forepaw somatosensory and motor networks, as well as in distant visual and parietal regions in the –Stim group (**C**). Additionally, mice recovering spontaneously exhibit increased anticorrelations between visual, somatosensory and motor regions (**C**). (**D**) Group-wise RSFC differences examined using unbiased spatial principle component analysis (PCA) of the group-level correlation difference matrix in panel C. First 2 PCs explain 70% of the variance between group differences at 4 weeks. Topography of PC1 reveals increased RSFC of –Stim mice within motor and sensory regions within both hemispheres with a notable island of increased connectivity within the remapped forepaw region. Pronounced anticorrelations can also be observed between posterior visual/retrosplenial regions and anterior sensorimotor regions. PC2 shows that surrounding sensory, medial motor, cingulate and more distant parietal regions also exhibit increased RSFC in –Stim mice compared to +Stim mice. Contours of original and remapped forepaw regions shown for reference. (**E**) The resulting eigenspectrum after PCA of the Week 4 correlation difference matrix in (**C**). PC1 at 4 weeks (green dot) explains 52% of the variance between groups at 4 weeks. (**F**) PCA-derived regions of interest (ROIs) for unbiased RSFC analysis. See *Figure 5—figure supplement 1* for ROI locations in each PC map. (**G**) 2-way repeated measures ANOVA (rmANOVA) shows brain wide differences in regional RSFC over time between groups. Regions having a significant group × time interaction (p<0.05 uncorrected for multiple comparisons) are shown. Notably, regions outside of the focal

*Figure 5 continued on next page*

*Figure 5 continued*

injury demonstrate significant differences in RSFC across groups. For example, RSFC between left secondary motor (M2) and distant retrosplenial and visual regions. Similarly, ipsilateral and bilateral RSFC in visual and retrosplenial cortices is also significantly altered between groups. (**H**) Left: stick and ball network diagram showing the effect of stimulation on recovery. Recovery for both groups was defined as the change in magnitude of ROI-based RSFC from 1 to 4 weeks (see *Figure 5—figure supplement 1*). The difference between these matrices (+Stim minus –Stim) determined the effect of contralesional excitation on spontaneous recovery. Significantly suppressed recovery was observed between nodes connected to the lesion and other regions (left primary sensory hindpaw/forepaw [S1hp/fp] and left visual; left M2 and left retrosplenial), between nodes distant to the site of direct injury (left and right visual cortex), and across hemispheres (right motor and left visual). Thicker lines indicate larger changes. Significant differences were those in panel having a significant group × time interaction as determined in panel G but corrected for multiple comparisons. Right: RSFC changes in control mice (n=9) over the experiment. No significant changes were observed. Statistical analyses were performed in MATLAB (2018a).

The online version of this article includes the following figure supplement(s) for figure 5:

**Figure supplement 1.** Regional resting-state functional connectivity (RSFC) between principle component analysis (PCA)-derived regions of interest (ROIs) and RSFC recovery.

**Figure supplement 2.** Recovery of functional connection density after stroke is suppressed by focal contralesional activity.

*supplement 2*) maps of global node degree for each group and time point were calculated as the number of functional connections exhibited by each pixel having a correlation coefficient z(r)>0.4. At baseline, both groups exhibit high node degree (reds) in motor, anterior somatosensory, medial parietal, ret, and visual regions, with fewer connections (blues) exhibited by lateral sensory areas (*Figure 5—figure supplement 2A*). Focal ischemia in left S1FP reduced node degree in local forepaw circuits, as well as lateral sensory and motor areas with only minimal differences evident across groups at week 1 (*Figure 5—figure supplement 2B*). However, the effects of chronic stimulation became profound by week 4. Substantial increases in node degree are evident in –Stim mice in several brain regions while +Stim mice exhibit topographically similar maps of node degree to those at week 1. The largest differences in global connection number between –Stim and +Stim mice at week 4 were observed within right forepaw cortex, large portions of motor cortex as well as posterior cingulate, medial parietal, retrosplenial, and visual areas (*Figure 5—figure supplement 2B*, week 4). Many of these regions exhibited a significant group × time interaction following a 2-way rmANOVA over the experiment (*Figure 5—figure supplement 2C*). To determine how chronic stimulation altered recovery of regional connection density, maps of node degree at week 1 were subtracted from those at week 4 in each group (*Figure 5—figure supplement 2D*). In –Stim mice, the number of functional connections increases over most of the cortex from week 1 to 4 (*Figure 5—figure supplement 2D*), whereas in +Stim mice, only portions of parietal and lateral somatosensory cortex exhibit any change. Group-wise differences in recovery (*Figure 5—figure supplement 2D*) reveal that chronic excitatory activity profoundly suppresses reestablishment of local and global connections within ipsilesional motor, small portions of visual and in the contralesional hemisphere, motor cortex, primary somatosensory forepaw and surrounding cortex, and visual cortices (*Figure 5—figure supplement 2E*).

## Focal contralesional excitation reduces global cortical activity

Suppression of local and global connectivity during recovery suggests that interventional optogenetic photostimulation imparts local and distant influences on cortical activity. In a separate experiment, we combined optogenetic photostimulation during awake OISI to visualize in real time the effects of contralesional excitation on global cortical activity. Similar to the above experiments, mice (n=16 CamK2a-ChR2 mice) were subject to photothrombosis of left S1FP, half of which received interventional optogenetic photostimulation beginning 1 day after photothrombosis (+Stim) while the other half recovered spontaneously (–Stim). Concurrent awake OISI during acute contralesional photostimulation was performed in +Stim and –Stim groups before, 1 and 4 weeks after photothrombosis. At baseline, all mice exhibited focal increases in right S1FP activity (reds) during photostimulation that spread to contralateral (left) S1FP and other motor regions approximately 8–10 s after stimulus onset (*Figure 6A*). While activity increases within the targeted circuit, subtle reductions in cortical activity can also be observed in surrounding non-targeted cortex. Thus, regional activity varies in and out of synchrony with the optogenetically driven circuit. As we have shown previously, correlating photo-stimulus evoked activity with all other brain pixels (i.e. effective connectivity mapping) sensitively captures this synchrony(*Bauer et al., 2018*; *Snyder and Bauer, 2019*). As an index of inhibition, we performed this procedure for all mice, and summed all pixels having a correlation coefficient less than

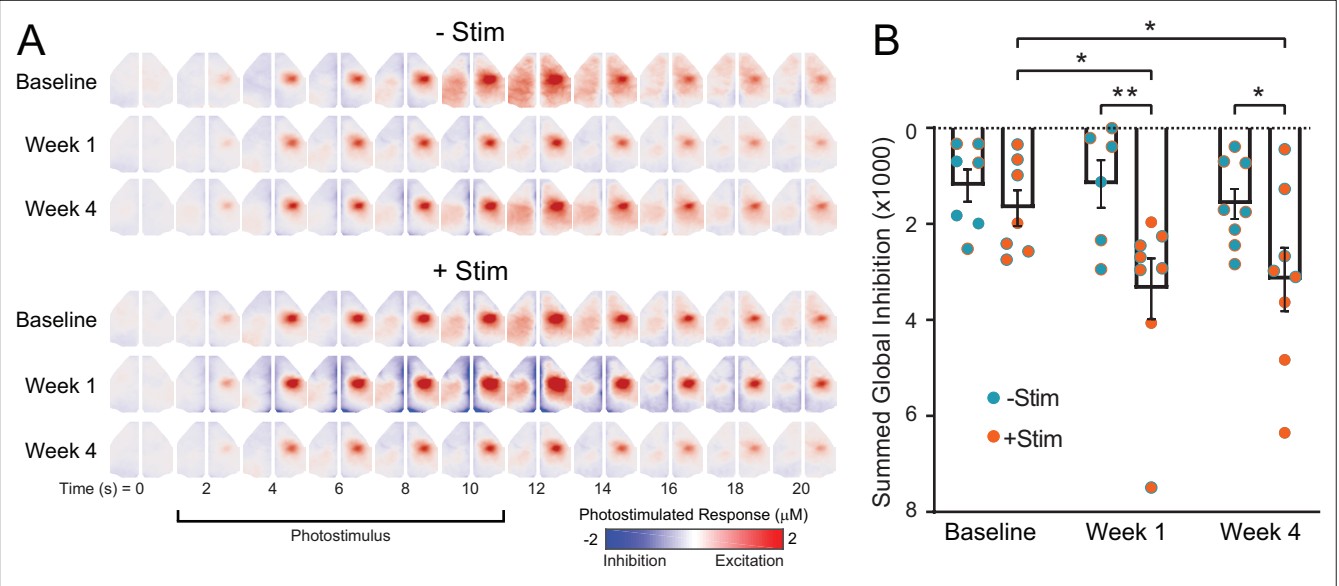

**Figure 6.** Contralesional photostimulation during awake optical intrinsic signal imaging (OISI) reveals global decreases in activity after stroke. Mice (+Stim: n=8, −Stim: n=8) were imaged before and after photothrombosis using awake OISI during photostimulation (473 nm, 0.5 mW, 10 Hz, 5 ms) delivered in a block design (10 s of photstimulation followed by 50 s of rest). (**A**) Group-averaged image sequences from 0 to 20 s. At baseline, all mice exhibited focal increases in right primary sensory forepaw (S1FP) activity (reds) during photostimulation that spread to contralateral (left) S1FP and other motor regions approximately 8–10 s after stimulus onset. Prior to activity increases within the targeted circuit, subtle reductions in cortical activity can be observed in contralateral cortex, including somatosensory, motor and more medial cortex at 4–8 s. (**B**) Quantification of activity reductions were performed by correlating photostimulus evoked time courses in each mouse with every other pixel, and summing all pixels having a correlation coefficient less than 0. Decreases in activity are markedly pronounced in +Stim mice at 1 and 4 weeks compared to baseline and were significantly larger in +Stim mice compared to −Stim mice. Linear mixed-effects model (LMM) revealed a main group effect ($F_{1,38}$=11.66, p=0.0015). All post hoc tests were performed as t-tests assuming unequal group variance and corrected for multiple comparisons using false discovery rate correction. Statistical analyses were performed in Graph Pad Prism 8. *=p<0.05; **=p<0.01; ***=p<0.001; ****=p<0.0001.

0. After photothrombosis these activity decreases are markedly pronounced in +Stim mice at 1 week compared to baseline (t(12)=2.4, p=0.02) which persist at 4 weeks (t(12)=2.1, p=0.04) (*Figure 6B*). Post stroke reductions of activity were significantly larger in +Stim mice compared to −Stim mice at both time points (1 week: t(12)=3.0, p=0.005); 4 weeks (t(12)=2.3, p=0.026) (*Figure 6B*).

## Contralesional activity inhibits expression of genes important for plasticity after stroke

The lack of local and global reconnection within the somatomotor network of +Stim mice suggests that contralesional stimulation might be affecting molecular mechanisms responsible for the formation of new connections after stroke. Using real time polymerase chain reaction (RT-PCR), we examined gene expression profiles of specific transcripts associated with inflammation, growth, inhibitory/excitatory signaling, synaptic plasticity, dendritic branching, and extracellular matrix remodeling following focal ischemia (*Figure 7*). Unsupervised hierarchical clustering was used to qualitatively examine broad changes in genetic expression in perilesional and contralesional tissue in both groups (*Figure 7A*). We identified three larger dendrograms (color coded orange, green, and black) segregating groups of genes based on patterns of expression, and independent of group or tissue assignment. Levels of expression in −Stim mice in either hemisphere provide a 'spontaneous recovery phenotype' (−Stim columns, *Figure 7A*) for evaluating against the effects of contralesional stimulation. The expression profile of the yellow dendrogram is characterized by opposite regulation in the ipsilateral (down regulation) and contralateral (up regulation) hemispheres of −Stim mice and encompass receptors and plasticity genes related to spines. Furthermore, in the black dendrogram genes weakly regulated by stroke in the ipsilateral hemisphere of −Stim mice displayed increased expression of growth factors and extracellular matrix genes by contralateral stimulation. In contrast, in the contralateral hemisphere of −Stim mice these genes were down regulated. The most pronounced differences in genetic expression between groups occurs within the green dendrogram. For example, in ipsilesional tissue, 27

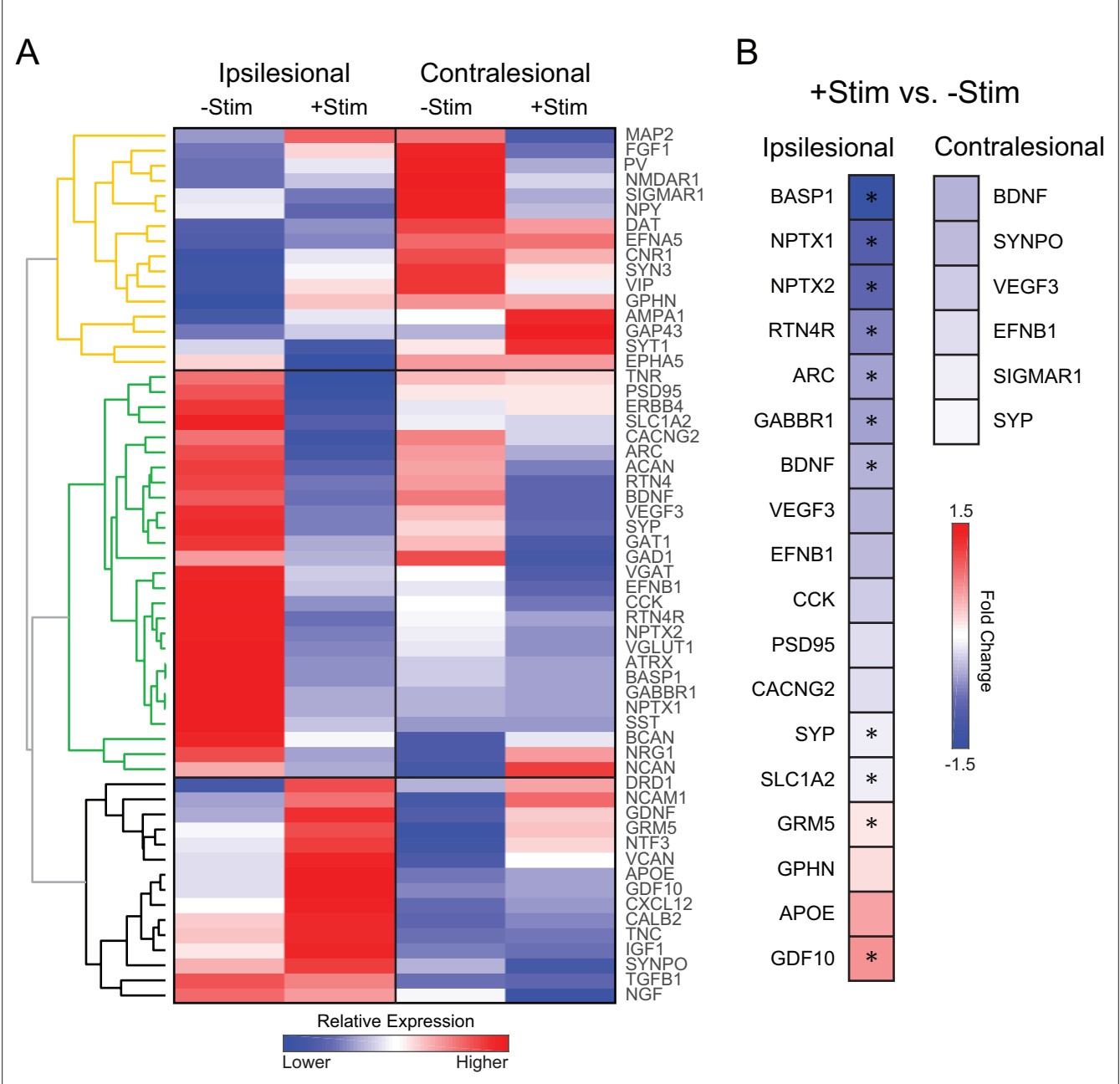

**Figure 7.** Contralesional activity alters expression of genes important for recovery quantitative real time polymerase chain reaction (RT-PCR) analysis was performed in perilesional and contralesional tissue of –Stim (n=5) and +Stim (n=5) mice. (**A**) Broad differences in genetic expression profiles were qualitatively examined with hierarchical clustering. Three larger dendrograms (color coded orange, green, and black) segregate groups of genes based on patterns of expression, independent of group or tissue assignment. Levels of expression in –Stim mice in either hemisphere provide a 'spontaneous recovery phenotype'. Within ipsilesional tissue, 27 genes upregulated (reds) in –Stim mice were all downregulated (blues) in +Stim mice (green tree). Mouse GAPDH was used as a normalization reference. (**B**) Fold change of expression in +Stim mice compared to –Stim mice in either perilesional or contralesional tissue. In panel B, colored boxes report fold change of any gene upregulated (reds) or downregulated (blues) with respect to –Stim mice with p≤0.05 (uncorrected). Starred boxes indicate significance following correction for multiple comparisons (p<0.1 following false discovery rate [FDR] correction). Contralesional stimulation significantly altered canonical pathways associated with plasticity after stroke. Statistical analyses were performed in Graph Pad Prism 8.

genes strongly upregulated (reds) in –Stim mice were all downregulated (blues) in +Stim mice (green tree, *Figure 7A and B*), while in the contralateral hemispheres most of these genes responded weakly to stroke.

To isolate the effects of contralesional stimulation after stroke, we compared expression levels in +Stim mice to those in –Stim mice in either hemispheres (*Figure 7B*). Colored boxes report fold change of any gene upregulated (reds) or downregulated (blues) with respect to –Stim mice with p≤0.05 (uncorrected). Starred boxes indicate significance following correction for multiple comparisons (p<0.1 following false discovery rate [FDR] correction). Contralesional stimulation significantly altered canonical pathways associated with plasticity after stroke. In perilesional tissue (top row, *Figure 7B*), this maneuver suppressed genes associated broadly with growth (BDNF), axonal guidance and regulation (BASP1 and RTN4), synaptic plasticity and regulation (ARC, SYP, NPTX1, and NPTX2), synaptic transmission (GABBR1), and increased expression of metabotropic glutamatergic receptor (GRM5) and an axonal growth promoter (GDF10).

## Discussion

Most stroke patients show some degree of spontaneous recovery, suggesting that endogenous mechanisms of repair are important. Thus, it is essential to understand what processes might inhibit or interfere with spontaneous repair mechanisms. Increased activity in homotopic brain regions contralateral to the infarct is associated with poorer recovery in patients after stroke. However, this observation is highly inconsistent across subjects and the mechanisms underlying poorer recovery are unknown. The present work examines how contralesional, excitatory activity in homotopic brain regions functionally connected to the site of injury influences cortical reorganization and global activity after stroke. We show that functional recovery occurs in tandem with formation of new cortical representations of the affected limb, restoration of RSFC within the affected somatosensory network, and more global renormalization of functional networks outside of the lesioned territory. Focal excitation of contralesional cortex significantly suppressed these processes, and altered transcriptional changes in several genes important for recovery.

### Better recovery is associated with more normalized patterns of activity in the affected circuit

Spontaneous recovery (i.e. in –Stim mice) was associated with new cortical representations of the right (affected) limb having an area of activation and response magnitude nearer to baseline levels (*Figure 2A and B*). Conversely, perilesional cortex in +Stim mice failed to remodel as completely and exhibited sustained reductions in evoked activity at week 4 (*Figure 2A and D*). Stimulation of the left (unaffected) limb resulted in the opposite picture. Responses in –Stim mice were indistinguishable from baseline responses while those in +Stim mice exhibited increased activity in right S1FP and right M1FP that persisted over the 4 week recovery period (*Figure 2C and D*, *Figure 3B*) consistent with disinhibition from the lesioned hemisphere (*Mohajerani et al., 2011*; *Shimizu et al., 2002*). The occurrence of new cortical S1FP representations in –Stim mice was therefore associated with more normal patterns of evoked activity in the larger S1FP circuit. Chronic contralesional excitatory stimulation prevented the formation of new cortical representations of the affected limb, and resulted in abnormally large representations of the unaffected limb. In parallel with cortical remapping, we also observed that the presence of new cortical S1FP representations in –Stim mice was associated with restoration of RSFC amongst functionally homotopic brain regions (*Figure 4D*, *Figure 4—figure supplement 1A*). That is, regions of homotopic synchrony with right S1FP shift from their original territory in left S1FP to remapped left S1FP representations. New representations also exhibit similar patterns of RSFC with other brain regions formally connected to the lesioned territory (e.g. left motor). Thus, more complete recovery experienced by –Stim mice was also associated with the reestablishment of more synchronous intrinsic activity within the larger somatomotor network.

The occurrence of new cortical S1FP representations along with a restoration of balanced stimulus evoked activity and homotopic RSFC in the S1FP circuit suggests that cortical remapping of disconnections after stroke provides a clue to how local circuits reconnect with brain networks to facilitate behavior. Following focal ischemia, peri-infarct regions appear to undergo large-scale changes in neuronal response properties. For example, in both human and animal studies peri-infarct regions

become more responsive to stimulation of somatomotor regions with which they are not typically associated (*Brown et al., 2009*; *Dijkhuizen et al., 2001*; *Jaillard et al., 2005*; *Nudo et al., 1996*; *Wei et al., 2001*). In rodents, peri-infarct remodeling correlates temporally with behavioral recovery (*Brown et al., 2009*; *Dijkhuizen et al., 2001*), and behavioral deficits in rats can be reinstated by ablating these remapped regions (*Castro-Alamancos and Borrel, 1995*), suggesting that remapped cortex assumes the function of brain regions lost to stroke. Re-establishing normal patterns of stimulus evoked or resting-state activity appears essential for regaining function. Brain networks returning toward more normal patterns of intrinsic organization after stroke (i.e. restored homotopic RSFC) also exhibit more normalized patterns of activation (*Brown et al., 2009*; *Dijkhuizen et al., 2001*; *Rehme et al., 2011*; *Grefkes et al., 2008*; *Allegra Mascaro et al., 2019*; *Kozlowski et al., 1996*). Abnormal intrinsic brain rhythms that give rise to abnormally strong patterns of task activation result in reduced variability of motor patterns after stroke (*Ramsey et al., 2016*). Abnormal brain activity could reflect an attempt to link, albeit inefficiently, regions that are disconnected, either by increasing neural activity upstream of the lesion (*Tennant et al., 2017*) or by rerouting activity through accessory regions (*Ramsey et al., 2016*; *Mohajerani et al., 2011*). Intermittent contralesional excitatory stimulation could exacerbate these processes.

We find that reconnection between remodeled cortex and homotopic brain regions parallels global network renormalization, this implies that these processes are both required for proper recovery. Recent mouse work probing causal relationships between local neural stimulation and recovery of specific (sensorimotor) behaviors further support the idea that more normal patterns of activity are associated with better behavior. For example, photostimulation of the ipsilateral primary motor cortex in transgenic Thy1-ChR2 mice promoted functional recovery and a return to normal patterns of optogenetically evoked perfusion changes in perilesional cortex (*Cheng et al., 2014*). In a separate study, optogenetic stimulation of thalamocortical axons also promoted the formation of new and stable thalamocortical synaptic boutons, which was correlated with enhanced recovery of somatosensory cortical circuit activity and forepaw sensorimotor performance (*Tennant et al., 2017*). Complementing these previous studies showing targeted photostimulation can improve recovery, our study adds nuance to the importance of site selection for therapeutic intervention by highlighting regions that suppress recovery.

## Increased contralesional activity negatively affects global network interactions

Ischemic stroke results in direct structural damage to local brain networks but also to functional damage to global networks outside of the lesioned territory (*Murphy and Corbett, 2009*; *Carrera and Tononi, 2014*; *Baldassarre et al., 2016*). While brain network dysfunction after stroke appears to be largely due to structural damage (*Griffis et al., 2019*; *Marebwa et al., 2017*; *Corbetta et al., 2015*; *Griffis et al., 2017*; *Saenger et al., 2018*), widespread functional disruptions are topographically linked within functional networks that can span across hemispheres (*Griffis et al., 2019*). Because functional outcome after stroke depends on the local site of ischemic injury as well as on remote connections (*Sinke et al., 2018*; *Straathof et al., 2019*; *Jones et al., 1996*), it is therefore necessary to consider more than just the ischemic territory when characterizing systems-level mechanisms of brain repair (*Grefkes and Fink, 2014*). ROI-based RSFC analysis of the somatomotor network shows clear changes in network synchrony over the 4 week recovery period in –Stim mice (top row, *Figure 4B*) that were not observed in +Stim mice (bottom row, *Figure 4B*). Beyond increased RSFC between right and remapped left S1FP at week 4, right S1FP exhibits higher RSFC with right M1FP and with regions anterior to the infarct in motor cortex. Supporting this qualitative observation, maps of node degree in –Stim mice indicate that large portions of motor and sensory cortices exhibit more functional connections than +Stim mice (*Figure 5—figure supplement 2*). Additionally, regions well outside of direct injury (e.g. visual, retrosplenial) also exhibit changes in functional connection number. These data suggest that both local and distant networks are altered after stroke, and that the degree to which these networks recover is impaired by increased contralesional stimulation (*Figure 5—figure supplement 2C*). However, these findings do not offer insights into the effect of increased contralesional activity on evolving internetwork interactions during recovery. To characterize global network interactions, we examined all pairwise RSFC over the cortex (*Figure 5*). Unexpectedly, several brain regions underwent dramatic changes in RSFC structure. In –Stim mice, parietal, visual,

and retrosplenial cortices all exhibited increased homotopic RSFC (*Figure 5C, D and G*) compared to +Stim mice. Furthermore, +Stim mice exhibited significant reductions in the magnitude of anticorrelations between functionally opposed networks residing within anterior (M1 and M2) and posterior (ret, V1p) cortices (*Figure 5B, D and G*). Dynamic patterns of anticorrelated activity segregate distinct processing streams across networks during both task and resting-state paradigms (*Fox et al., 2005*; *Greicius et al., 2003*). In humans, the most prominent example occurs between the dorsal attention network and default mode network. Altered synchrony across large cortical distances (>several mm), as observed by reduced anticorrelated activity in +Stim mice at week 4, could prevent inter-network communication necessary for normal function. Reduced node degree in +Stim mice might indicate network-level inefficiencies, exemplified by more asynchronous intrinsic activity within these networks. We have recently shown (*Hakon et al., 2018*) that improved tactile proprioception after stroke was associated with increased node degree in motor, somatosensory, and parietal cortices in mice, all regions relevant for processing proprioception and touch (*Carey et al., 2016*). Reduced within network connectivity (i.e. global node degree) combined with poorer interactions across networks (i.e. reduced anticorrelations) could impair multi-domain behavioral output in mice after stroke.

## Contralesional stimulation alters transcriptional changes in genes important for recovery

Recovery from stroke is associated with neuronal and synaptic plasticity, the formation of new structural and functional connections that take over those lost due to infarction. These changes are likely to occur both locally (at the peri-infarct region) and at distant sites where changes in connectivity might occur (e.g. contralesional hemisphere). We examined changes in the expression of select genes associated with neuronal plasticity to determine if contralesional stimulation altered expression of genes associated with spontaneous recovery (*Figure 7*). While changes in mRNA do not necessarily lead to proportional translation into protein, regulated transcripts do reflect potential involvement of the corresponding proteins in the regulation (activation or inhibition) of cellular processes in response to injury, plasticity, and/or intervention that could affect brain function. While distinct phases of plasticity occur over different time scales (*Wieloch and Nikolich, 2006*), good recovery of the peri-infarct is associated with upregulation of neurotrophic signaling (*Stowe et al., 2007*), growth-associated gene expression (*Carmichael et al., 2005*), and markers of axonal formation/migration (*Benowitz and Carmichael, 2010*) and synaptogenesis (*Murphy and Corbett, 2009*; *Kraft et al., 2018*). Some of these repair-related molecular changes are also observed in homologous, contralesional sites (*Cramer and Riley, 2008*).

Our cumulative results suggest that spontaneous recovery following focal ischemia may be suppressed by chronic contralesional excitation. We used cluster analysis to determine if changes in neuroplasticity genes associated with spontaneous recovery (−Stim group) were suppressed by chronic contralesional excitation (*Figure 7A*). Four groups were entered into the cluster analysis (ipsilesional and contralesional tissue in both groups). Hierarchical clustering revealed three distinct clusters that segregated patterns of gene expression. Functional recovery following small cortical lesions preferentially occurs in ipsilateral hemisphere (*Biernaskie et al., 2005*; *Shanina et al., 2006*). Of particular interest were a collection of genes that demonstrated significant increased expression in ipsilesional cortical tissue in the −Stim group, but were suppressed in the +Stim group: BASP1, NPTX1, NPTX2, RTN4R, ARC, BDNF, SYP, and SLC1A2 (*Figure 7B*). In line with these findings, we recently showed that remapping of forepaw cortex is dependent on activity-dependent synaptogenesis, and that blocking ARC-dependent remapping limited behavioral recovery (*Kraft et al., 2018*). NPTX1 and 2 mediate synaptic clustering of AMPA receptors, are implicated in synapse formation and maintenance (*Xu et al., 2003*; *Pribiag and Stellwagen, 2014*) and facilitate metabotropic glutamate receptor-mediated long-term depression (*Cho et al., 2008*). Increased excitatory input from the contralesional hemisphere chronically after stroke appears to depress these repair-related molecular events.

Contralateral stimulation markedly downregulates genes coding for proteins involved in axonal sprouting. RTN4R forms part of a signaling complex (i.e. Nogo) that inhibits axonal growth and limits experience-dependent neural plasticity (*Schwab and Strittmatter, 2014*) and BASP1 a growth-associated protein, that plays critical roles in synaptic plasticity and neurite outgrowth (*Korshunova et al., 2008*). In the post stroke axonal sprouting transcriptome, GDF10 is one of the most highly

upregulated genes during the initiation of axonal sprouting in peri-infarct cortical neurons (*Li et al., 2015*). Despite a lack of remapping in +Stim mice, this group exhibited significantly increased expression of GDF10. New axonal growth and subsequent navigation likely require coordination amongst several molecular cues to ensure proper synaptic targeting.

Altering cell signaling, or modulating inhibitory or excitatory tone can profoundly affect recovery (*Clarkson et al., 2010*; *Clarkson et al., 2011*; *Kokinovic and Medini, 2018*). The negative effect of contralesional excitation on recovery is thought to be due to increased perilesional inhibitory influences. While we did survey for markers of inhibition and excitation (*Supplementary file 1*), most were not observed to be significantly altered between groups after correcting for multiple comparisons. However, stimulated mice did exhibit significant reduction in ipsilesional expression of GABABR1, a G protein-coupled receptor abundantly localized to glutamatergic synapses which regulates excitatory neurotransmission and synaptic plasticity (*Ainsworth et al., 2016*). Alone, focal ischemia results a loss of GABA B-mediated interhemispheric synaptic inhibition in stroke periphery (*Kokinovic and Medini, 2018*). Sustained activation of glutamate receptors following focal ischemia leads to rapid downregulation of GABA B receptors via lysosomal degradation (*Zemoura et al., 2016*; *Maier et al., 2010*). Stroke induced degradation of GABAB1 receptors could lead to the observed compensatory upregulation of GABBR1 transcript prevented by contralateral stimulation. Glutamate-induced downregulation of GABABR1 could lead to reduced inhibition of glutamatergic synapses, increasing synaptic excitability potentially to the point of increased excitotoxicity (*Zemoura et al., 2016*). While we did not observe group-wise differences in lesion volume, future studies are required to determine whether increasing ipsilesional inhibition during the recovery phase (*Hiu et al., 2016*, *Balbi et al., 2021*) offsets the effects of contralesional excitation. It is also important to note, that the current analysis was performed at a single time point (14 days after stroke), during dynamically changing molecular environment (*Wieloch and Nikolich, 2006*).

## Interhemispheric influences on recovery and implications to clinical stroke

In one model of information exchange across hemispheres, excitatory information from one hemisphere conveyed by the corpus callosum activates inhibitory cells in the opposite hemisphere, thus inhibiting activity (*Palmer et al., 2012*; *Petrus et al., 2019*). This inhibition between the left and right hemispheres may be important for bilateral dexterity in upper and lower limb movements, and is required for integrating bilateral sensory and motor signals (*Shuler et al., 2002*; *Petrus et al., 2020*). Unilateral injury (e.g. stroke) modifies interhemispheric connectivity in such a way that disrupts the balance across hemispheres by decreasing the inhibitory influence of the affected hemisphere (*Shimizu et al., 2002*; *Xerri et al., 2014*; *Carmichael, 2012*). Non-invasive brain stimulation techniques, such as rTMS and tDCS (Transcranial Direct-Current Stimulation), have been used to influence rehabilitation after stroke by affecting this (Excitatory-Inhibitory) E/I balance. However, evidence supporting the clinical use of these techniques has been inconsistent. Some clinical trials have shown that these methods have a large, positive effect on motor recovery (*Adeyemo et al., 2012*), while other studies were inconclusive (*Hao et al., 2013*). A portion of the variability in outcome is due to grouping patient populations with different types of strokes (ischemic vs hemorrhagic) and/or with lesions in different locations (cortical vs subcortical). Furthermore, the lack of refinement for spatial and cellular specificity, limits the use of rTMS and tDCS for examining specific populations of neurons or connections important for recovery.

Utilizing optogenetic targeting, we examine the role of contralesional, homotopic excitation on functional brain organization after stroke, a maneuver that might mimic the increased activity occurring during use of the unaffected limb. Sustained, increased activation of contralesional M1 using rTMS or through the use of the 'good limb' is associated with poorer clinical outcome (*Turton et al., 1996*; *Feydy et al., 2002*), findings corroborated in rodent models and in other, non-motor systems (*Boonzaier et al., 2018*; *Allred et al., 2005*; *Allred et al., 2010*). Our results indicate that even brief epochs of contralesional activity (3 min/day) during the repair phase after stroke is detrimental to recovery, possibly by extending an inhibitory influence well beyond the affected circuit (*Figure 6*) and critical time-windows. Further, while acute imbalances in interhemispheric balance could be transient, chronic excitation could extend the period of imbalance. Increased global inhibition might interfere with local and distant network communication, which might result in the lack of reconnection of both

local circuits lost due to stroke, and the subsequent reintegration of these circuits into global resting-state networks. It is important to point out that the role of the ipsilesional vs contralesional hemispheres on stroke recovery depends on the size and location of the infarct for review see *Keith et al., 2017* and timing of rehabilitation can profoundly alter outcome (*Kozlowski et al., 1996*; *AVERT Trial Collaboration group, 2015*). Because structural and functional asymmetries persist in well-recovered stroke patients even after clinical symptoms have normalized (*Julkunen et al., 2016*), more systematic approaches are required to unravel causal influences for differences in outcome.

## Limitations and future work

The design of the present study was intended to examine how early, chronic stimulation affected cortical remodeling after stroke. The duration of the experiment did not include time points when spontaneously recovering mice have been shown to exhibit more complete recovery, e.g. out to 8 weeks or longer (*Brown et al., 2009*). However, at 4 weeks, we see a clear divergence in patterns of local and global circuit repair. Our assessments of cortical remodeling and network reorganization were based on systems-level, hemoglobin-contrast-based measures of brain activity. Visualizing more direct measures of neural activity, e.g. through the use of genetically encoded calcium indicators (*Gu et al., 2018*; *Brier et al., 2019*), or voltage sensitive dyes (*Mohajerani et al., 2011*), would allow for examining evolving neural network interactions as they relate to behavior (*Cramer et al., 2019*) or evolving microvascular function (*Lee et al., 2016*; *Sakadžić et al., 2015*, *Sunil et al., 2020*). Several unanswered questions remain regarding the choice of stimulation parameters, sites targeted, and neuronal subpopulations involved. For example, are the negative effects of increased contralesional activity most pronounced during a particular time window after injury? Would focal stimulation of cortical regions functionally distinct from the lesioned circuit (e.g. contralesional visual cortex) result in a similar negative influence on recovery? Are there specific excitatory or inhibitory subpopulations of neurons that have a particular influence on plasticity? These are exciting questions that we are currently exploring. Further work is also needed to disambiguate circuit specific roles that mediate the negative affect of contralesional stimulation on recovery. Right S1FP connects widely to several other cortical (e.g. S1, M1, and M2) and subcortical structures (e.g. striatum, thalamus, and midbrain, superior colliculus, and cerebellum) involved in integrating sensory information to coordinate motor output (*Oh et al., 2014*). It is therefore possible that contralateral stimulation is impairing recovery, not through a direct effect of monosynaptic projections from contralesional to perilesional tissue, but by indirect modulation of cortical output (*Quiquempoix et al., 2018*) or through descending projections. Future studies aimed at mapping brain wide activity (e.g. via fMRI or diffuse optical tomography [*Reisman et al., 2017*]) following focal stimulation would help disambiguate a circuit specific role on recovery, and determine whether activity in the site stimulated is the main contributor to outcome. The limited scope of our RT-PCR analysis provided associations between expression of select neuroplasticity genes and systems level changes across groups post stroke. More complete evaluations of the post stroke transcriptome, and confirmation as to which genes and proteins are altered following contralesional stimulation will allow for a more complete understanding of how this maneuver affects cortical remodeling after stroke.

## Materials and methods
### Mice

A total of 105 adult male mice expressing channelrhodopsin (ChR2) under the mouse calcium/calmodulin-dependent protein kinase II alpha (*CamK2a*) promoter were used for experimentation. CamK2a-ChR2 mice were generated using Cre-Lox recombination (parent strains: CamK2a-Cre, B6.Cg-Tg(Camk2a-cre)T29-1Stl/J, stock number: 005359; Lox-ChR2, B6;129S-Gt(ROSA)26Sortm32(CAG-COP4*H134R/EYFP)Hze/J, stock number: 012569, The Jackson Laboratory). Mice were housed in enriched environment cages and given food and water ad libitum with a 12 hr on:12 hr off light cycle. Around 94 mice were used for stroke recovery experiments, and 9 mice were used as controls (see below and *Figure 1*). About 10 mice were euthanized 2 days after photothrombosis for infarct volume quantification. About 10 mice (5 −Stim, 5 +Stim) were euthanized 14 days after photothrombosis for qPCR. 35 mice (15 −Stim, 20 +Stim) were used for behavioral testing. The OISI data were collected on the remaining mice for remapping and RSFC experiments (9 controls, 10 −Stim, 15 +Stim) or for

visualizing cortical activity during optogenetic photostimulation before and after stroke (8 −Stim, 8 +Stim). All experimental protocols were in compliance with the Institutional Animal Care and Use Committee at Washington University in St. Louis.

## Enriched environment

Enriched housing has been shown to improve functional recovery after stroke (*Quattromani et al., 2018*; *Madinier et al., 2014*), thus increasing the dynamic range of assessments of functional recovery. Mice were housed in 24" × 17" × 8" cages (Nalgene) for 1 month prior to and during the entirety of the experiment. Cages contained: Nestlets, Mouse Arch, Mouse Huts, Mouse Igloos, Mouse Tunnels, Fast Trac (bio-serv), and Enviro-dri crinkle paper. These components were re-arranged or replaced every 1–2 weeks to provide new stimuli for the mice.

## Animal surgery

Prior to imaging, a clear Plexiglas window was placed on the exposed skull as per our previous studies (*Wright et al., 2017*). A single injection of buprenorphine-SR (1.0 mg/kg SC) was given 1 hr prior to the surgery. Mice were anesthetized with 3% isoflurane inhalation in air for induction, and kept at 1.5% for the duration of the surgery. A clear Plexiglas window with predrilled holes, sterilized with Chlorhexidine and rinsed with sterile saline, was secured to the scalp with dental cement (C&B-Metabond, Parkell Inc, New York, USA). Mice were monitored for 2 days post-surgery. No imaging was performed during this post-operative period.

## Photothrombosis

Focal ischemia was induced via photothrombosis as previously described (*Kraft et al., 2018*). Under isoflurane anesthesia (3.0% induction, 1.5% maintenance), mice were placed in a stereotactic frame. A 532 nm green DPSS laser (Shanghai Laser & Optics Century) collimated to a 0.5 mm spot was centered on left S1FP (0.5 mm anterior to bregma, 2.2 mm left of bregma) at low power (<0.25 mW). The laser was turned off, and mice were then given 200 µL Rose Bengal dissolved in saline (10 g/L) via intraperitoneal injection (i.p.) injection. After 4 min, the laser power was set to 23 mW and illuminated left S1FP for 10 min.

## Interventional optogenetic photostimulation

A randomized subset of mice received chronic, intermittent optogenetic photostimulation of homotopic S1FP in the right hemisphere for 5 consecutive days/week for 4 weeks beginning 1 day after photothrombosis. Awake mice were head-restrained in a custom stereotactic frame using the pre-tapped holes in the cranial windows mounted to the skull, with legs supported by a suspended, freely rotating foam ball. While mounted, mice are able to stand with legs supported, walk, whisk, groom, etc. The output of a fiber-coupled 473 nm blue DPSS laser (Shanghai Laser & Optics Century) was mounted to a stereotactic micromanipulator arm collimated to a 0.5 mm spot and centered on right S1FP (0.5 mm anterior to bregma, 2.2 mm right of bregma) using a stereotactic micromanipulator. Prior to interventional photostimulation, laser power was adjusted between 0.2 and 1 mW and set to a level just below that which elicited overt behavioral responses (e.g. forepaw or whisker motor movements in sync with stimuli) as determined visually by the person delivering the photostimulus. Interventional photostimuli consisted of 20 ms pulses delivered at 10 Hz for 1 min on, 3 min off, 1 min on, 3 min off, 1 min on (3 min/day) as per a previous study reporting functional improvement when this stimulus was delivered to perilesional tissue (*Cheng et al., 2014*). Control mice were not subject to photothrombosis, but received interventional optogenetic photostimulation. –Stim mice were subject to photothrombosis, but not interventional photostimulation.

## Electrical forepaw stimulation

To evoke cortical activity in S1FP, transcutaneous electrical stimulation was applied to the left and right forepaws as previously described (*Kraft et al., 2018*) by placing microvacular clips (Roboz) on either side of the wrists. Electrical stimulation was provided in a block design (AM Systems Model 2100) with the following parameters: 5 s rest, 10 s stimulation (0.5 mA, 0.3 ms duration, 3 Hz) followed by 35 s of rest as previously described (*Bauer et al., 2014*). About 15 min of data were collected for each paw (18 trials per paw total).

## Optical intrinsic signal imaging

All OISI occurred before, and 1 and 4 weeks after photothrombosis.

Mapping evolving RSFC and cortical responses to electrical peripheral stimulation: sequential illumination was provided by a custom ring of LEDs centered at four wavelengths (478 nm, 588 nm, 610 nm, and 625 nm). A cooled, frame-transfer EMCCD camera (iXon 897, Andor Technologies) captured diffuse light reflectance from the skull over a field-of-view of approximately 1 cm². Data were binned 4 × 4 on camera, resulting in a frame rate of 120 Hz (hemodynamic imaging at 30 Hz). Hardware was controlled via a data acquisition card (PCI 6733, National Instruments, Texas, USA) and computer (custom-built, SuperLogics, Massachusetts, USA) using custom-written software (MATLAB 2018a). Mice were anesthetized by i.p. of a ketamine/xylazine cocktail (86.9 mg/kg ketamine, 13.4 mg/kg xylazine). To facilitate longer imaging times, after the initial bolus, mice were infused subcutaneously with a saline-ketamine cocktail (34.8 mg/kg/hr ketamine) during the imaging sessions as previously described (*Bauer et al., 2014*). A heating pad kept at 37°C maintained the mouse body temperature. The 30 min of activation data (15 min per paw, 18 stimulus presentations per paw) and up to 45 min of resting-state data were collected for each mouse in 5 min datasets (75 min of data total per mouse).

Mapping the effects of contralesional excitation on global cortical activity: for experiments consisting of optogenetic photostimulation and awake OISI, the sequential LED illumination described above was interleaved with the photostimulation laser as per our previous reports (; *Lee et al., 2021*). A 473 nm notch filter (Chroma, ZET405/473-488/NIRm) positioned between the lens and camera prevented reflected laser light from saturating the camera's sensor. Image acquisition rate was 50 Hz (10 Hz hemodynamic imaging). Optogenetic photostimuli were delivered to right S1FP in a block design for 30 min (473 nm, 0.5 mW, 20 ms pulses delivered at 10 Hz for 10 s followed by 50 s of rest repeated 30 times).

## Image processing

All image processing and analysis was performed using MATLAB 2018a (Mathworks).

### Initial data reduction

Data from all mice were subject to an initial quality check prior to spectroscopic analysis as per our previous reports (*Bero et al., 2012*; *Wright et al., 2017*). Raw reflectance was converted to changes in hemoglobin (Hb) concentration at each pixel and each time point as we have previously reported (*Bero et al., 2012*; *Wright et al., 2017*). Each pixel's time series was downsampled from 30 to 1 Hz, and data were filtered between 0.009 and 0.08 Hz for RSFC analysis, and 0.009–0.5Hz for task based measures. Global signal regression was performed prior to evaluating RSFC and cortical responses to electrical forepaw stimulation. All imaging data were affine-transformed to a common atlas space determined by the positions of the junction between the coronal vessel separating the olfactory bulb from the cortex and sagittal suture along midline and lambda as we have done previously (*Bero et al., 2012*; *Wright et al., 2017*).

### Mapping responses to peripheral forepaw stimulation

Unless otherwise stated, total hemoglobin was used as contrast because it offers the highest contrast-to-noise (*Bauer et al., 2018*), is more spatially specific than oxygenated or deoxygenated Hb (*Ramsey et al., 2016*; *Bauer et al., 2018*) and is most closely linked to underlying neural activity (*Sirotin et al., 2009*). For each stimulation block, baseline images (1–5 s before stimulation) were averaged together and subtracted from the all images in the block. Stimulation blocks for each paw were averaged together. Maps of peak responses were calculated by averaging *Figure 2* seconds before and 2 s after stimulation offset and used for activation area and magnitude calculations. Because evoked responses at early time points post stroke can be difficult to detect, group-averaged maps of peak responses at baseline were thresholded at 75% of maximum to define a threshold for what constituted a response for all mice at all time points. The maximum response above this threshold was used for activation magnitude calculations, and the number of pixels above this threshold was used for the area calculations. All magnitude and area values were then normalized by the mean baseline values for each group. Activation incidence images were created by binarizing all pixels above the 75% threshold for each mouse and time point. To calculate the map of Cohen's D at 4 weeks post stroke, evoked

response T-maps were created across stimulus blocks for each mouse. All pixels having a T-value >2 were included.

## Resting-state functional connectivity analysis

ROIs for left and right forepaw (original or remapped) were determined by group-averaged activation maps from the baseline and 4 week time points and all pixels within the thresholded activations (see above) were averaged to create time courses for each ROI. Forepaw motor representations were determined by stimulation experiments at week 4. The ROI traces were correlated with all other time traces in the shared brain mask to create maps of RSFC for the local forepaw somatosensory circuit. Global RSFC circuits were evaluated via zero-lag correlation for all pixel pairs within the shared brain mask for each mouse. Spatial PCA was performed on the group-averaged, whole cortex correlation difference matrix at 4 weeks to evaluate the largest sources of variance between the –Stim and +Stim groups as we have done (*Kraft et al., 2020*). Following extensive permutation resampling (see statistics), the first 2 PCs were determined significant and used to define ROIs for RSFC analysis. Maps of PC1 and 2 were symmetrized about midline, smoothed and thresholded at 85% of maximum for positive and negative values resulting in 7 ROIs within each hemisphere (*Kraft et al., 2020*). Maps of global node degree were calculated as described previously (*Hakon et al., 2018*; *Quattromani et al., 2018*) by thresholding whole cortex correlation matrices at z(r)≥0.4. Connections above this threshold were set to 1, and summed over both hemispheres to produce a binarized measure of global node degree for all pixels within the shared brain mask.

## Mapping responses to photostimulus-evoked excitatory activity

Image sequences of photostimulus evoked activity of total hemoglobin were block averaged and baseline subtracted as described above. Block averaged time courses in the photostimulated region (right S1FP) were created by averaging all time courses within 4 pixels (310 µm) of the targeted site and correlated with all time courses in the field of view. As we have demonstrated previously , these correlation maps reveal patterns of activity in and out of synchrony with the optogenetically driven region. To create a measure of global inhibition, all pixels having a correlation coefficient below 0 were summed for each mouse at each time point.

## Infarct quantification

Mice (n=5 –Stim and n=5 +Stim) were deeply anesthetized with FatalPlus (Vortech Pharmaceuticals, Michigan, USA) and transcardially perfused with heparinized PBS. The brains were removed and fixed in 4% paraformaldehyde for 24 hr and transferred to 30% sucrose in PBS. After brains were saturated, they were snap-frozen on dry ice and coronally sectioned (50 µm thick slices spaced 300 µm apart) on a sliding microtome. After mounting, slices were allowed to dry for 24 hr and stained with cresyl violet, and cover slipped. Stained sections were imaged with a NanoZoomer under bright field setting and infarct area was quantified in ImageJ for each slice. Infarct volume was calculated as total infarct area × 300 µm. Statistical analyses were performed using Graph Pad Prism 8.

## Brain extraction and RT-PCR

Mice (n=5 –Stim and n=5 +Stim) were deeply anesthetized with isoflurane and transcardially perfused with 0.01 M PBS. Brains were extracted, placed in a brain matrix, and coronal sections spanning from –0.5 mm posterior to bregma to 1 mm anterior to bregma were collected. Based on the approximate location of S1FP from the Paxinos atlas, tissue from the perilesional and contralesional regions were dissected, snap-frozen on dry ice, and stored at –80°C. Total RNA was extracted using RNeasy Mini Kit (Qiagen) and reverse transcribed with the cDNA Reverse Transcription kit (Invitrogen). Quantitative PCR was performed with SYBR Green using the ABI QuantStudio 12 K Flex system in the default thermal cycling mode. Mouse GAPDH was used as a normalization reference. Relative mRNA levels were calculated using the comparative Ct method. Prior to clustering, expression levels of each gene were mean subtracted across groups and normalized to unit variance. RT-PCR analyses were performed using Microsoft Excel 2016 and MATLAB 2018a. Genes examined and primers used are tabulated in *Supplementary file 1*.

## Cylinder rearing

Cylinder rearing recording and analysis was done as previously described (*Baskin et al., 2003*; *Kraft et al., 2018*). Briefly, mice receiving photothrombosis were placed in a 1000 mL glass beaker and recorded for 5 min. A blinded observer manually analyzed videos to determine the amount of time that the (1) right paw, (2) left paw, or (3) both paws made contact with the glass walls. Paw-use asymmetry was calculated as (% left paw use – % right paw use)/(% left paw use + % right paw use). Mice having paw-use asymmetry at baseline greater than 50% for either limb were discarded from any subsequent analysis. Statistical analyses were performed using Graph Pad Prism 8.

## Statistical analyses

Statistical analyses were performed using either Graph Pad Prism 8 or Matlab 2018a. All Pearson-R values were converted to Fisher-Z values prior to statistical testing. Differences between stroke groups for imaging (e.g. pairwise RSFC) and behavioral measures were analyzed using 2-way rmANOVA. In the control group, differences across time were assessed using 1-way rmANOVA. For limited cases where data points from individual mice were missing, significant effects were analyzed using a LMM fit with a restricted maximum likelihood function. Equal variability of differences (i.e. sphericity) was not assumed, and degrees of freedom were adjusted using the Geisser and Greenhouse correction. Significant effects were evaluated post hoc; these and other pairwise comparisons were performed using a 2-tailed Student's t-test assuming unequal group variance and corrected for multiple comparisons by controlling the FDR. Maps of statistical differences were corrected on a clusterwise basis. For all above tests, significance was achieved if (corrected) $p < 0.05$ except for RT-PCR ($p < 0.1$). The number of PCs used for data-driven ROI selection were determined from group-averaged, whole cortex correlation matrices. Extensive permutation resampling (3000 iterations) between groups was used to determine the amount of variance expected in the first two eigenvalues of the PCA decomposition in the null case. In the true eigenspectrum, PCs 1 and 2 (>70% variance explained) exceeded the 90th percentile and were considered statistically significant.

## Acknowledgements

This work was supported by National Institute of Health grants R01-NS102870 (AQB), K25-NS083754 (AQB), R37NS110699 (JML), R01NS084028 (JML), R01NS094692 (JML), R01NS078223 (JPC), P01NS080675 (JPC), R01NS099429(JPC), F31NS089135 (AWK), and F31NS103275 (ZPR), the McDonnell Center for Systems Neuroscience (AQB), The Alborada Trust (TW), The Wachtmeister Foundation (TW) and The Swedish Research Council (TW). We also thank Grant Baxter and Jasmine Park for their assistance with imaging data acquisition, and Shannon Macauley for helpful discussions on reporting RT-PCR data.

## Additional information

### Funding

| Funder | Grant reference number | Author |
| --- | --- | --- |
| National Institutes of Health | R01NS102870 | Adam Q Bauer |
| National Institutes of Health | K25NS083754 | Adam Q Bauer |
| National Institutes of Health | R37NS110699 | Jin-Moo Lee |
| National Institutes of Health | R01NS084028 | Jin-Moo Lee |
| National Institutes of Health | R01NS094692 | Jin-Moo Lee |
| National Institutes of Health | R01NS078223 | Joseph P Culver |

| Funder | Grant reference number | Author |
|---|---|---|
| National Institutes of Health | P01NS080675 | Joseph P Culver |
| National Institutes of Health | R01NS099429 | Joseph P Culver |
| National Institutes of Health | F31NS089135 | Andrew W Kraft |
| National Institutes of Health | F31NS103275 | Zachary Pollack Rosenthal |
| McDonnell Center for Systems Neuroscience | | Adam Q Bauer |
| Alborada Trust | | Tadeusz Wieloch |
| The Wachtmeister Foundation | | Tadeusz Wieloch |
| Vetenskapsrådet | | Tadeusz Wieloch |

The funders had no role in study design, data collection and interpretation, or the decision to submit the work for publication.

## Author contributions

Annie R Bice, Data curation, Formal analysis, Investigation, Methodology, Software, Visualization, Writing - original draft, Writing - review and editing; Qingli Xiao, Data curation, Formal analysis, Investigation, Methodology; Justin Kong, Formal analysis; Ping Yan, Formal analysis, Methodology; Zachary Pollack Rosenthal, Data curation, Methodology, Writing - review and editing; Andrew W Kraft, Conceptualization, Investigation, Methodology; Karen P Smith, Formal analysis, Investigation, Methodology; Tadeusz Wieloch, Conceptualization, Data curation, Visualization, Writing - review and editing; Jin-Moo Lee, Conceptualization, Data curation, Funding acquisition, Investigation, Project administration, Resources, Supervision, Writing - review and editing; Joseph P Culver, Conceptualization, Funding acquisition, Resources, Supervision; Adam Q Bauer, Conceptualization, Formal analysis, Funding acquisition, Project administration, Resources, Supervision, Writing - original draft, Writing - review and editing

## Author ORCIDs

Zachary Pollack Rosenthal http://orcid.org/0000-0001-8787-0858
Andrew W Kraft http://orcid.org/0000-0002-5168-3986
Adam Q Bauer http://orcid.org/0000-0002-8364-3209

## Ethics

All procedures described below were approved by the Washington University Animal Studies Committee in compliance with the American Association for Accreditation of Laboratory Animal Care guidelines (Protocol #20-0022).

## Decision letter and Author response

Decision letter https://doi.org/10.7554/eLife.68852.sa1
Author response https://doi.org/10.7554/eLife.68852.sa2

# Additional files

## Supplementary files

• Supplementary file 1. Genes surveyed for RT-PCR Analysis.
• Transparent reporting form

## Data availability

Data reported in Figures 1, 6, 7 are publicly available on figshare: Figure 1, Figure 6, Figure 7. Data reported in Figures 2, 3, 4, 5 are unavailable due to technical issues with storage hard drives. Analysis code is available on GitHub (copy archived at swh:1:rev:5845aae7c056dbc293600c9aae87a0d170a00ff1).

The following datasets were generated:

| Author(s) | Year | Dataset title | Dataset URL | Database and Identifier |
|-----------|------|---------------|-------------|-------------------------|
| Bauer A | 2022 | Cylinder Rearing Scores | https://doi.org/10.6084/m9.figshare.19773487.v1 | figshare, 10.6084/m9.figshare.19773487.v1 |
| Bauer A | 2022 | Neuroimaging Data Pre/Post Stroke for 26-03-2021-RA-eLife-68852 | https://doi.org/10.6084/m9.figshare.19773244.v1 | figshare, 10.6084/m9.figshare.19773244.v1 |
| Bauer A | 2022 | RT-PCR Data | https://doi.org/10.6084/m9.figshare.19773364.v1 | figshare, 10.6084/m9.figshare.19773364.v1 |

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
