## [Editor Report]

Overall, Bice et al., present new work using optogenetics-based stimulation to test how this affects stroke recovery in mice. The study provides interesting evidence that this stimulation may be harmful, and not helpful, and opens new avenues for both basic and therapeutic research.

---

## [Decision Letter]

**Decision letter after peer review:**

Thank you for submitting your article “Homotopic contralesional excitation suppresses spontaneous circuit repair and global network reconnections following ischemic stroke” for consideration by *eLife*. Your article has been reviewed by 2 peer reviewers, one of whom is a member of our Board of Reviewing Editors, and the evaluation has been overseen by Tamar Makin as the Senior Editor. The reviewers have opted to remain anonymous.

Essential revisions:

1. The studies in the manuscript utilize brain stimulation, and use it as a goal of modelling behavioral limb use. In particular, the optogenetic stimulation protocol of primary somatosensory forelimb cortex is said to mimic overuse of the limb. It is not clear how stimulation of a subset of neurons in somatosensory cortex mimics limb overuse. Further, this stimulation is, in totality, very brief. Limb overuse does not appear to align with this brain stimulation protocol.

2. The parameters of the stimulation are set and not altered in these studies. They were chosen based on Cheng et al., (ref 25). This is interesting because in this publication, the optical stimulation parameters, delivered into peri-stroke cortex, produced recovery. Would different stimulation parameters have a different outcome on recovery or functional connectivity? Or is this contralateral stimulation site the main determinant of the negative effect on these two? The stimulation setup is very different in this Cheng et al., study, apparently to the present study. However, this is not clear as the actual stimulation is not described. In Cheng et al., the stimulation was specifically to layer V pyramidal neurons with an optrode in Thy1-ChR2 mice. A cranial window is described in the methods section, but this appears to be for imaging. How were the ChR2 neurons stimulated in this present study?

3. Relative to the last issue in #2, if the stimulation was done with an implanted optrode, what was done with control: the -stim condition. Was this an implanted optrode but not activated? Or, if stimulation was done with an LCD on the skull or a window, was this done in -stim?

4. There is substantial and ever-deeper analysis of the rsMRI data (Figures 4-6). This is all supportive of the overall core finding, that contralateral stimulation of neurons with ChR2 in this protocol impairs recovery. But each successive level of rsMRI analysis does not really add a new amount of evidence-this array of figures is not a new or independent set of data.

5. The behavioral readout is not well documented and is a key readout for the main claims in the paper. For example, in the methods; "The laser power ranged between 0.2mW – 1mW and was set to a level just below that which elicited overt behavioral output (e.g. forepaw or whisker motor movements in sync with stimuli)." for example, how is this measured? By eye? Quantitatively with tracking approaches? Also, the cylinder test methods notes there is only 1 blinded scorer, but how consistent is this person? Why not use a machine vision approach (if the videos are saved, I strongly suggest this is quantified differently, or if they do the new experiments suggested, this is used on the new video data).

*Reviewer #2 (Recommendations for the authors):*

Additional studies that might define the biology of recovery here, is to more completely flesh out the interaction of stimulation with recovery, so as to better prove that the site of stimulation is the main negative effect.

---

## [Author Response]

Essential revisions:1. The studies in the manuscript utilize brain stimulation, and use it as a goal of modeling behavioral limb use. In particular, the optogenetic stimulation protocol of primary somatosensory forelimb cortex is said to mimic overuse of the limb. It is not clear how stimulation of a subset of neurons in somatosensory cortex mimics limb overuse. Further, this stimulation is, in totality, very brief. Limb overuse does not appear to align with this brain stimulation protocol.

We appreciate the reviewer’s comment and have removed the statement “a maneuver mimicking the use of the unaffected limb” from the Abstract, Introduction, and Study Design (Results). We have also modified the Abstract, Introduction (pages 2,3) and Discussion (pages 12 and 15) to more clearly articulate why we chose our model.

2. The parameters of the stimulation are set and not altered in these studies. They were chosen based on Cheng et al., (ref 25). This is interesting because in this publication, the optical stimulation parameters, delivered into peri-stroke cortex, produced recovery. Would different stimulation parameters have a different outcome on recovery or functional connectivity? Or is this contralateral stimulation site the main determinant of the negative effect on these two?

The reviewer is correct that we chose the stimulation paradigm implemented precisely because of its efficacy in the Cheng, et al., study. It is possible, and likely, that other stimulation parameters would result in improved or worsened recovery, depending on whether the ipsilesional or contralesional hemisphere was targeted. These are exciting questions that we are currently exploring, but feel an exhaustive exploration of which stimuli/networks are best/worst to target is beyond the scope of the current manuscript. We have added text to the limitations section of the Discussion on page 16 to address these remaining questions.

The stimulation setup is very different in this Cheng et al., study, apparently to the present study. However, this is not clear as the actual stimulation is not described. In Cheng et al., the stimulation was specifically to layer V pyramidal neurons with an optrode in Thy1-ChR2 mice. A cranial window is described in the methods section, but this appears to be for imaging. How were the ChR2 neurons stimulated in this present study?

We have clarified the “Interventional Optogenetic Photostimulation” section of the Methods (page 17) to more accurately describe our photostimulation protocol.

3. Relative to the last issue in #2, if the stimulation was done with an implanted optrode, what was done with control: the -stim condition. Was this an implanted optrode but not activated? Or, if stimulation was done with an LCD on the skull or a window, was this done in -stim?

We have also clarified on page 17 of the methods that control mice were not subject to photothrombosis, but received interventional optogenetic photostimulation. -Stim mice were subject to photothrombosis, but not interventional photostimulation.

4. There is substantial and ever-deeper analysis of the rsMRI data (Figures 4-6). This is all supportive of the overall core finding, that contralateral stimulation of neurons with ChR2 in this protocol impairs recovery. But each successive level of rsMRI analysis does not really add a new amount of evidence-this array of figures is not a new or independent set of data.

The reviewer is correct that original Figures 4-6 report different results from the same set of functional neuroimaging data. Each set of analyses is intended to examine unique aspects of either local or global network organization after stroke. Figure 4 probes the effect of contralesional activity on local forepaw somatosensory circuit recovery using functional maps (task-evoked forepaw representations) as ROIs. The data-driven analyses in Figure 5 are agnostic to the experimental paradigm to determine whether targeted contralesional activity affects global network interactions (following or in the absence of focal ischemia). Because recovery of local and distant functional connection strength after stroke (node degree, quantified in original Figure 6) depends on the RSFC data in Figures 4 and 5, results from these analyses might be anticipated. We have therefore decided to move this Figure to supplemental material (new Figure 5—figure supplement 2).

New independent data are included in the manuscript as new Figure 6, where we show the effect of optogenetic photostimulation before and after stroke on global cortical activity in awake animals. For additional information, please see our response to concern #2 by Reviewer #2 below.

5. The behavioral readout is not well documented and is a key readout for the main claims in the paper. For example, in the methods; “The laser power ranged between 0.2mW – 1mW and was set to a level just below that which elicited overt behavioral output (e.g. forepaw or whisker motor movements in sync with stimuli).” For example, how is this measured? By eye? Quantitatively with tracking approaches?

We have clarified in response to Essential Point #2.2 above and in the “Interventional Optogenetic Photostimulation” section of the Methods (page 17) that overt behavioral responses to the interventional photostimulation were determined visually.

Also, the cylinder test methods notes there is only 1 blinded scorer, but how consistent is this person? Why not use a machine vision approach (if the videos are saved, I strongly suggest this is quantified differently, or if they do the new experiments suggested, this is used on the new video data).

We agree with the reviewer that a more efficient scoring method less prone to scorer-to-scorer variability would be optimal. Prior to examining the cylinder rearing data set in the current paper, the blinded scorer (J.K.) was previously trained by A.R.B. and A.W.K. using datasets previously published in Kraft et al., Science Translational Medicine, 2018. After a period of initial training, scorers are instructed to evaluate baseline (pre-stroke) rearing data. Scoring consistency is evaluated through (1) data-driven determination of mouse handedness (approximately 20% of mice are left- or –right handed, defined in our study as >50% use asymmetry) which removes those mice from subsequent evaluation, and (2) statistical evaluation between the results of the trainee and those previously tabulated by A.R.B and A.W.K from the Kraft et al., study. While we are confident that our scoring protocol yields consistent results across scorers, manual scoring is time consuming. To that end, we are currently working on a machine vision approach to scoring, but this methodology is not currently ready for application.

Reviewer #2 (Recommendations for the authors):Additional studies that might define the biology of recovery here, is to more completely flesh out the interaction of stimulation with recovery, so as to better prove that the site of stimulation is the main negative effect.

We agree and have expanded on this important point in the Discussion on pages 15-16.